# Retrieval of ozone profiles from OMPS limb scattering observations

Carlo Arosio[1], Alexei Rozanov[1], Elizaveta Malinina[1], Kai-Uwe Eichmann[1], Thomas von Clarmann[2], and John P. Burrows[1]

[1]Insitute of Environmental Physics, University of Bremen, Bremen, Germany
[2]Karlsruhe Institute of Technology, Karlsruhe, Germany

*Correspondence to:* carloarosio@iup.physik.uni-bremen.de

**Abstract.** This study describes a retrieval algorithm developed at the University of Bremen to obtain vertical profiles of ozone from limb observations performed by the Ozone Mapper and Profiler Suite (OMPS). This algorithm is based on the technique originally developed for use with data from the SCanning Imaging Absorption spectroMeter for Atmospheric CHartographY (SCIAMACHY) instrument. As both instruments make limb measurements of the scattered solar radiation in the Ultraviolet
(UV) and Visible (Vis) spectral ranges, an underlying objective of the study is to obtain consolidated and consistent ozone profiles from the two satellites and to produce a combined data set. The retrieval algorithm uses radiances in the UV and Vis wavelength ranges normalized to the radiance at an upper tangent height to obtain ozone concentrations in the altitude range of 12–60 km. Measurements at altitudes contaminated by clouds in the instrument field of view are identified and filtered out. An independent aerosol retrieval is performed beforehand and its results are used to account for the stratospheric aerosol load in
the ozone inversion. The typical vertical resolution of the retrieved profiles varies from $\sim 2.5$ km at lower altitudes ($< 30$ km) to $\sim 1.5$ km about 45 km and becomes coarser at upper altitudes. The retrieval errors resulting from the measurement noise are estimated to be 1–4 % above 25 km, increasing to 10–30 % in the upper troposphere. OMPS data are processed for the whole year 2016. Results are compared with the NASA product and validated against profiles derived from passive satellite observations, or measured in situ by balloon-borne sondes. Between 20 and 60 km, OMPS ozone profiles typically agree with
data from the Microwave Limb Sounder (MLS) v4.2 within 5–10 %, whereas in the lower altitude range the bias becomes larger, especially in the tropics. The comparison of OMPS profiles with ozonesonde measurements shows differences within $\pm 5$ % between 13 and 30 km at northern mid- and high-latitudes. At southern mid- and high-latitudes, an agreement within 5–7 % is also achieved in the same altitude range. An unexpected bias of approximately 10–20 % is detected in the lower tropical stratosphere. The processing of the 2013 data set using the same retrieval settings and its validation against ozonesondes reveals
a much smaller bias; a possible reason for this behavior is discussed.

## 1   Introduction

Ozone is one of the most important trace gases in the atmosphere. It is most abundant in the stratospheric 'ozone layer', which absorbs strong ultraviolet (UV) radiation, heating this atmospheric region and acting as a protective layer against biologically harmful radiation. It plays a crucial role in the radiative budget of the stratosphere, determines the stratospheric temperature
profile and impacts on atmospheric circulation and climate. Because of its relevance to both science and society, ozone-related

researches expanded after the discovery of the springtime ozone hole in Antarctica and the subsequent recognition that man-made release of chlorofluorocarbon compounds depletes the stratospheric ozone layer (Molina and Rowland, 1974; Farman et al., 1985). Although nowadays the stratospheric ozone chemistry is generally well understood, there are still several issues to be clarified. These are related to the expected ozone recovery after the adoption of the Montreal protocol, stratospheric

circulation and temperature responses to the increase of greenhouse gases (Li et al., 2009) as well as long term ozone trends. For example, Solomon et al. (2016) focused the attention on the Antarctic region, investigating possible signatures of an ozone healing. Analyzing observations collected each September since 2000, the authors suggested that the fingerprints of an ozone recovery can be identified in both the increase of its column amount and in the decrease of the areal extent of the ozone hole. The issues related to changes in the Brewer Dobson Circulation (BDC), possibly linked to climate changes, have

been investigated by several studies, which consider the ozone concentration in the lower stratosphere a good proxy to track changes in the stratospheric circulation. Among them, Aschmann et al. (2014) used combined ozone time series from satellite instruments and ozonesondes to investigate changes in BDC after the beginning of the century and identified an asymmetry in the northern and southern branches. Stiller et al. (2017) suggested a shift of the subtropical mixing barriers as an explanation.

For all these kinds of studies, reliable long-term data sets are needed from both ground-based and satellite instruments.

Recent attempts to consistently merge a large number of different data sets into long-term time series are reported by Froidevaux et al. (2015) and Davis et al. (2016) both including also other species than ozone. Steinbrecht et al. (2017) and Sofieva et al. (2017) focused on ozone trends, revealing a global statistically significant increase in its amount after 2000 above 35 km. Other authors, as Kyrölä et al. (2013), Eckert et al. (2014), Gebhardt et al. (2014) and Nedoluha et al. (2015) pointed out an unexpected decadal negative trend in the ozone abundance in the upper tropical stratosphere.

During the last few decades, several remote sensing observation techniques have been used to derive ozone concentrations from the troposphere up to the mesosphere (Hassler et al., 2014). Following the birth of the space age, instrumentation of different kinds began to be developed. Space-borne remote sensing measurements in the Ultraviolet-Visible (UV-Vis) spectral range have traditionally been of two types: nadir viewing and solar occultation spectrometers; the former instruments point downward and are characterized by a good horizontal coverage, whereas the latter look directly into the solar disk, featuring

a good vertical resolution and a strong signal. The limb sounding technique, widely used by more recent satellite instruments, combines the advantage of these two: the long path through the atmosphere provides a high sensitivity to trace gases and the variation of the observation angle enables a better vertical resolution with respect to the nadir geometry, featuring a much higher horizontal sampling as compared to the occultation measurements. Limb observation geometry has also been used to measure scattered solar radiance and/or atmospheric emission in the InfraRed (IR) and microwave spectral regions. Using the

scattered solar light, measurements during daylight only are possible, whereas, using the emission signatures, observations can be performed during both day and night. The accuracy/sensitivity of limb measurements decreases with the altitude in the lower stratosphere and troposphere, as the increasing optical thickness along the line of sight leads to a saturation of the measured signal. The presence of clouds in the field of view acts as an additional limitation.

The limb scatter technique was for the first time successfully exploited by the LORE/SOLSE (Limb Ozone Retrieval Ex-

periment/Shuttle Ozone Limb Sounding Experiment) instrument launched in 1997 by NASA. Two instruments followed this

mission: the Optical Spectrograph and Infrared Imager System (OSIRIS) launched in February 2001 (Llewellyn et al., 1997) and the SCanning Imaging Absorption spectroMeter for Atmospheric CHartographY (SCIAMACHY), launched in March 2002 (Burrows et al., 1995; Gottwald and Bovensmann, 2011). SCIAMACHY made observations in the UV, Vis, Near InfraRed (NIR) and Short Wave InfraRed (SWIR) spectral ranges till April 2012, when the platform-to-ground communication was lost. A few aging satellite instruments, such as OSIRIS and the Microwave Limb Sounder (MLS), are still operating, contributing to the task of continuous monitoring the stratospheric ozone. At the end of 2011, just a few months before the end of ENVISAT lifetime, the Ozone Mapping and Profiler Suite (OMPS) instrument was launched on board the Suomi-National Polar-Orbiting Partnership (SNPP) platform and it is still operational (Flynn et al., 2014). The spacecraft has a nominal 13:30 local time ascending sun-synchronous orbit and flies at a mean altitude of 833 km. Scientific data collection started at the beginning of 2012. OMPS comprises three instruments: the Nadir Mapper, Nadir Profiler and Limb Profiler (LP). Only the latter is of interest for our study (see Flynn et al., 2014, for a review of the full suite).

After the launch of the satellite, the NASA team developed a retrieval chain to derive ozone profiles and many by-products from OMPS limb observations, which are publicly available. Besides, at the University of Saskatchewan a 2-D geometry retrieval has been applied to OMPS-LP measurements (Zawada et al., 2017).

This paper presents ozone profile retrievals from OMPS-LP observations performed at the University of Bremen. The algorithm we use was developed based on the SCIAMACHY v3.0 ozone retrieval (Jia et al., 2015). As the two instruments have a very different spectral resolution and the measurement techniques differ in many respects, e.g. in terms of spectral channels, wavelength ranges, atmospheric/scene sampling and radiance collection, a direct application of the SCIAMACHY retrieval scheme to OMPS-LP measurements was not possible. Although the algorithm presented in this manuscript has been newly developed starting from the one used to process SCIAMACHY data, the same radiative transfer model, a similar retrieval approach and the same spectroscopic and atmospheric parameters databases were used to minimize the systematic errors between the data sets. The underlying objective of the study is the creation of a consolidated product and the merging of the OMPS-LP and the SCIAMACHY time series, in order to obtain a long-term continuous data set. In Sect. 2, the OMPS instrument is introduced: its geometry of observation, relevant characteristics and issues related to the retrieval of ozone are briefly discussed. The third section is focused on the retrieval methodology, starting with a general description of the inversion algorithm used in this work. A more detailed characterization of the retrieval procedure follows, including the applied cloud filter and the approach to consider aerosol extinction profiles. Sect. 4 presents at first a comparison with NASA ozone profile retrieval algorithm; then MLS and ozonesonde data sets are used for a validation of our product. Main results, remaining issues and possible future improvements are addressed in the conclusions.

## 2    OMPS-LP instrument

### 2.1    General features and main issues

The main objective of OMPS-LP is to monitor the ozone vertical distribution within the Earth middle atmosphere at high accuracy level. It images the Earth atmosphere by viewing its edge (limb) from space. The closest approach of the sensor line

of sight to the Earth surface is referred to as Tangent Point (TP) and the altitude of this point above the Earth geoid is called Tangent Height (TH); the limb geometry is schematically drawn in the left panel of Fig. 1. The OMPS-LP sensor views at the Earth limb backwards with respect to the flying direction, through three vertical slits: the central one is aligned along the nadir track, whereas the other two are cross-track. As shown in the right panel of Fig. 1, the TPs are located about 25° latitude South of the sub-satellite point. The spacecraft completes 14–15 orbits per day and the instrument performs normally 180 limb observations (referred to as states) per orbit.

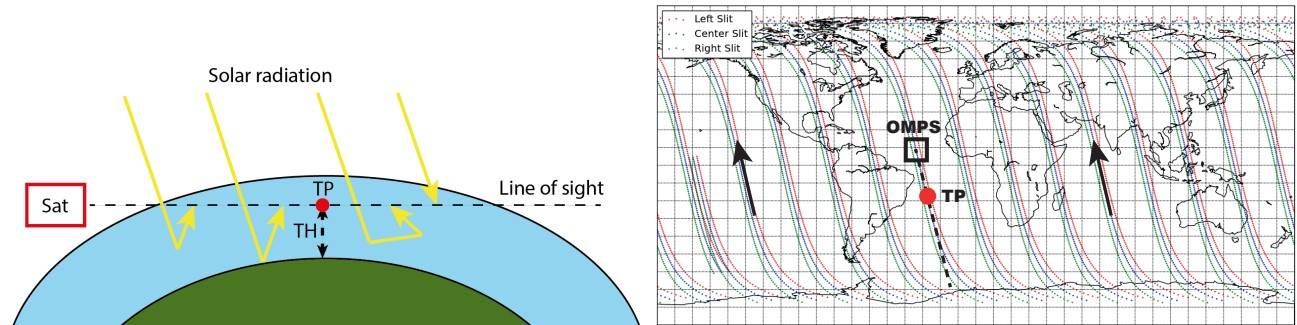

**Figure 1.** Left panel: schematic diagram of the viewing geometry of a satellite limb observation, showing the so called Tangent Point (TP) and its height above the Earth, or Tangent Height (TH). Right panel (adapted from Bhartia et al., 2013): OMPS daily orbits and observation geometry sketch; black arrows indicate the satellite flight direction and the red dot approximately locates the Tangent Point (TP).

OMPS-LP measures limb scattered radiance in the spectral range of 280–1000 nm. A particular characteristic of this instrument is the use of a prism spectrometer instead of a grating disperser. The employed prism provides a spectral resolution that degrades with wavelength, from 1 nm in the UV region up to 40 nm in the NIR. OMPS-LP observes the full altitude range at the same time, without vertical scanning, and radiance is collected by means of a Charge-Coupled Device (CCD). Each slit covers a vertical range of 112 km, the instantaneous vertical field of view of each detector pixel is about 1.5 km and the vertical sampling is 1 km at TP (Jaross et al., 2014). The use of the CCD detector poses a great challenge as regards the dynamic range: indeed, due to the decrease of the atmospheric density, scattered solar radiation from the Earth limb decreases by at least five orders of magnitude along the considered vertical range. Therefore, in order to cover the required dynamic range, four images at a 2-D physical CCD are taken for each slit: the full atmosphere is imaged at two integration times and through a large and a small aperture (Jaross et al., 2014). Ground processing is then needed to select unsaturated signals and combine down-linked pixels from different images in a single radiance file, that is finally re-sampled and mapped onto a regular grid. Left panel of Fig. 2 shows examples of radiance profile, displaying the large dynamic range of measured values, whereas the right panel depicts examples of spectral signal-to-noise ratio (SNR) at different altitudes. Jumps in this plot are related to the switch between the sampled images: for example, the jump between large and small aperture that occur at 450 nm (fixed threshold). In the retrieval scheme we take care not to consider spectral ranges crossing this fixed boundary.

One of the most important issues that affects the quality of the limb scattering technique is the TH registration. The required high pointing accuracy cannot be directly reached for OMPS-LP sensor, because the star-tracker on board the SNPP satellite

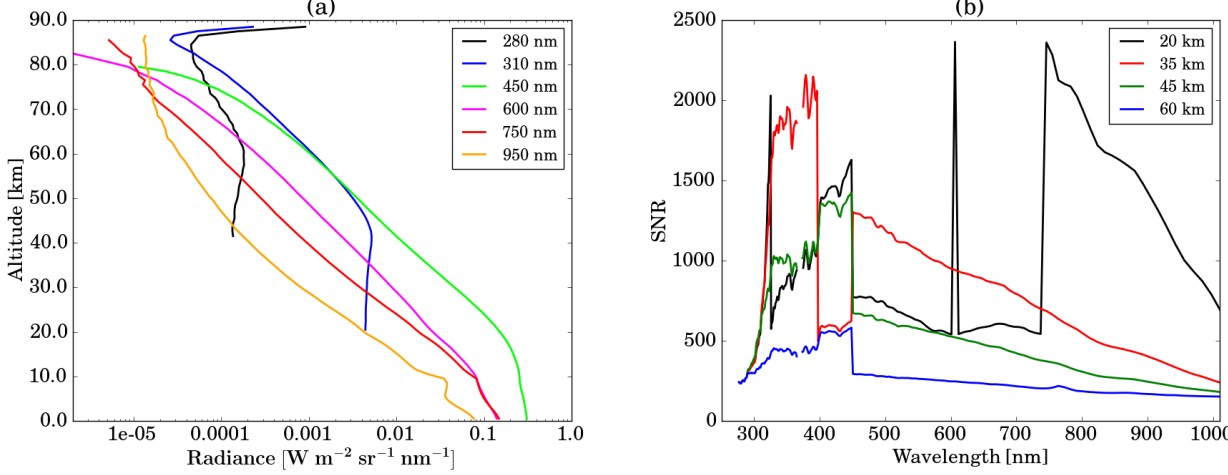

**Figure 2.** Left panel: example of OMPS-LP radiance profiles at some selected wavelengths; right panel: OMPS-LP signal-to-noise ratio (SNR) at different tangent heights.

is mounted on a distant position from the instrument, so that thermal effects and mis-alignments of the instrument focal plane play an important role.

To solve this problem, several pointing corrections are established during the level 1b gridded (L1G) data processing, as described in Moy et al. (2017):

5    1. Fixed adjustments between 1 and 2 km, depending on the slit.

2. TH variation related to the heating up of the instrument, as it approaches northern mid-latitudes.

3. Dynamic TH variation within each orbit, characterized by an almost linear dependence with state number.

While the first two are implemented in the current v2.5 version of L1G data, a satisfactory explanation for the latter variation still has to be found and this effect is currently not accounted for. However, following NASA recommendations in (DeLand 10 et al., 2017), we implemented a linear TH adjustment as a function of latitude, with values ranging from +300 m at the South Pole till -100 m at the North Pole.

The second important issue that affects the accuracy of the limb radiance is the so-called stray light. The general phenomenon of stray light describes photons that are registered by the detector at wavelengths or altitudes which they do not belong to. For example, with multiple images on a single detector, photons from the IR part of one slit can be scattered into the UV part of the 15 neighboring image. This problem was reduced with both a thorough study of the point spread function during the pre-launch operations and the careful application of cutoff filters at the focal plane (Jaross et al., 2014). Stray light is mainly an issue at high altitudes, with levels that are usually less than 10 % of the measured value.

Transient events can affect the instrument reliability: energetic charged particle can penetrate through the CCD shielding and cause transients in pixel signal. Such events are frequent in the so-called South Atlantic anomaly.

## 2.2 OMPS-LP observation geometry

Several angular coordinates are needed in the retrieval algorithm to correctly describe the observation geometry; satellite azimuth ($\varphi$), solar azimuth ($\varphi_0$) and solar zenith angle ($\psi_0$) at the TP are reported for three THs (25, 35 and 45 km) in the L1G data files and are used to define the geometry of the observation. The solar zenith angle ($\psi_0$) is defined as the angle between the local vertical at the TP and the sun pointing vector. The azimuth angles ($\varphi$ and $\varphi_0$) are defined as the angles between the direction to North Pole and the projections of the solar beam and the instrument line of sight, respectively, on the plane orthogonal to the normal vector at the TP.

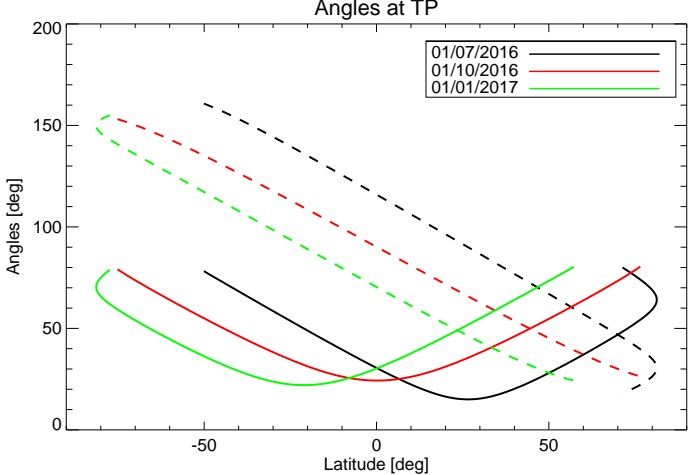

**Figure 3.** Solar zenith angles (solid lines) and scattering angles (dashed lines) at the TP along three OMPS orbits on the following dates: Jul 1, 2016, Oct 1, 2016 and Jan 1, 2017.

Combining azimuth and zenith angles, the scattering angle $\theta$ at the TP can be computed as:

$$\cos(\theta) = \sin(\psi_0)\cos(\varphi - \varphi_0) \tag{1}$$

This is an important quantity that defines the scattering geometry. In Fig. 3 values of scattering angles together with solar zenith angles are plotted as a function of latitude for three OMPS orbits in different seasons. Solar zenith angles are shown as solid lines, with symmetric values with respect to the equatorial region, whereas scattering angles are plotted as dashed lines. Only solar zenith angles less than 80° are plotted and the ozone retrieval is run only for the corresponding states, usually 140 per orbit, to avoid high stray light levels. The latitude coverage in different seasons can be assessed from the figure.

## 3 Retrieval method

### 3.1 Theoretical basis

The retrieval of ozone profiles is performed using the regularized inversion technique with the first order Tikhonov constraints (Tikhonov, 1963; Rodgers, 2000). The non-linearity of the inverse problem is accounted for using an iterative approach. The forward modeling takes into consideration atmospheric multiple scattering in the framework of the approximate spherical solver of the SCIATRAN radiative transfer model (Rozanov et al., 2014). Thereby, the CDI (Combined Differential-Integral) approach is employed to solve the radiative transfer equation: first, the entire radiation field is calculated in the pseudo-spherical approximation for a set of solar zenith angles using the finite difference method. Pseudo-spherical approximation means that the direct solar beam is traced in a fully spherical geometry while a plane parallel atmosphere is assumed to calculate the scattered light. Then, an integration along the line-of-sight is carried out in a spherical geometry, i.e. intersecting a spherical shell atmosphere, accounting also for the atmospheric refraction. Thereby the single scattering contribution is calculated fully-spherically while the multiple scattering contribution at each point along the line of sight is approximated by an angular integration of the pseudo-spherical radiative field calculated at the first step (Rozanov et al., 2000). Weighting functions are calculated using the same method as for the radiance, but considering only the single scattering contribution.

Linearizing the forward model around an initial guess state $\boldsymbol{x_0}$, the general equation that has to be solved can be written as:

$$\boldsymbol{y} = \boldsymbol{y_0} + \mathbf{K}(\boldsymbol{x} - \boldsymbol{x_0}) + \boldsymbol{\epsilon} \tag{2}$$

where $\boldsymbol{y}$ is the measurement vector, $\boldsymbol{y_0}$ is the simulated spectrum, $\mathbf{K}$ is the linearized forward model operator represented by the weighting function matrix, $\boldsymbol{x}$ is the state vector and $\epsilon$ represents errors of any kind. Following (Rodgers, 2000), the solution of Eq. (2) can be estimated iteratively. Taking into consideration that in our algorithm the retrieval is performed from a zero a priori profile, the iterative step $i+1$ can be expressed as:

$$\boldsymbol{x_{i+1}} = (\mathbf{K_i^T} \mathbf{S}_\epsilon^{-1} \mathbf{K_i} + \mathbf{S_0} + \mathbf{S_1^T} \gamma \mathbf{S_1})^{-1} \mathbf{K_i^T} \mathbf{S}_\epsilon^{-1} (\boldsymbol{y} - \boldsymbol{y_i} + \mathbf{K_i} \boldsymbol{x_i}) \tag{3}$$

Here, $\mathbf{S}_\epsilon$ is the measurement noise covariance matrix. $\mathbf{S_0}$ is the diagonal matrix optimized to constrain the solution within physically meaningful values and minimize a possible negative bias caused by the use of a zero a priori profile. The effect of the chosen matrix is significant only at tropical low altitudes and globally at high altitudes, where the ozone concentration is very small. Finally, $\mathbf{S_1}$ is the first order derivative matrix ($\mathbf{S_1^T} \gamma \mathbf{S_1}$ is the first order Tikhonov term). The diagonal matrix $\gamma$ contains altitude dependent weights, used to constrain the smoothness of the retrieved profile. In the following, the sum $\mathbf{S_0} + \mathbf{S_1^T} \gamma \mathbf{S_1}$ will be named as $\mathbf{S_r}$.

### 3.2 Algorithm implementation

For the ozone vertical profile retrieval from OMPS-LP, four spectral segments are selected: three in the UV spectral region (Hartley and Huggins bands) and one in the visible range (Chappuis band); the former ranges are sensitive to the upper stratospheric ozone, whereas the latter to the lower stratospheric region, where the peak of the number density occurs. In order

to avoid strong absorption bands of water vapor and $O_2$, wavelengths in the ranges 580.0–607.0 nm and 620.0–635.0 nm are rejected. A complete treatment of these absorption features requires line-by-line calculations, that are computationally expensive. The altitude range over which the retrieval is performed spans between 12 and 60 km above the sea level. The vertical grid is fixed throughout the processing and covers the retrieval range at evenly spaced steps of 1 km. To prepare the measurement vector, limb radiance in each spectral interval is normalized with respect to a limb measurement at an upper TH, in order to provide a self calibration of the instrument and reduce the effect of surface/cloud reflectance. In addition, for longer wavelength intervals, a polynomial is subtracted from the logarithm of the normalized radiance in order to remove slowly variable spectral features, e.g. caused by Rayleigh or aerosol scattering (Rozanov et al., 2011). Equation (4) explicitly shows the measurement vector at the j-th TH and details about spectral segments and TH normalizations are listed in Table 1. The last column provides the information about the subtracted polynomial in the measurement vector: first order in the visible range, zeroth order or no polynomial in the UV region.

$$y_j = \log\left(\frac{I_{TH_j}}{I_{TH_{norm}}}\right) - P_n \tag{4}$$

**Table 1.** List of the spectral segments considered for the ozone retrieval with corresponding altitude ranges, THs used for the normalization and the order of the subtracted polynomial ( - means that no polynomial is subtracted).

| Altitude range [km] | Spectral segment [nm] | Normalization TH [km] | Polynomial order |
|---|---|---|---|
| 46-60 | 285-300 | 63.5 | - |
| 35-46 | 305-313 | 52.5 | - |
| 31-36 | 322-331 | 47.5 | 0 |
| 12-33 | 508-660 † | 42.5 | 1 |

† 580.0–607.0 and 620.0–635.0 nm ranges are rejected.

In the forward model, the radiation is calculated taking into account $O_3$, $NO_2$ and $O_4$, which have spectral signatures in the selected spectral ranges. Cross sections of these gases are taken from Serdyuchenko et al. (2014), Bogumil et al. (2000) and Hermans (2011), respectively. Cross sections are beforehand convolved to the OMPS-LP spectral resolution. Ancillary pressure and temperature profiles are taken from the Global Modeling and Assimilation Office (GMAO) interpolated data set, provided by the NASA team together with OMPS-LP L1G radiances.

Before the main retrieval procedure, a shift and squeeze correction is applied in the Chappuis band to the modeled spectrum with respect to the measured one. This pre-processing is performed for each observation at each TH independently and is introduced to account for issues related to the spectral calibration and possible thermal expansion of the detector. Typical values for the spectral shift are inside the range [+1,+4] nm for the first point of the interval and [-2,+1] nm for the last spectral point. Due to the relatively low spectral resolution of the instrument, the differential absorption structure in the Huggins band is largely smoothed out and the UV retrieval uses either normalized radiances themselves or their slopes. As a consequence, the influence of a possible spectral misalignment is rather small and the shift and squeeze algorithm in not applied in the UV.

In the pre-processing procedure, we obtain the $\mathbf{S}_\epsilon$ matrix from the fit residuals, fitting absorption features of all relevant gases in the selected spectral windows.

The inversion scheme is then iteratively run employing the Eq. (3). The state vector $\boldsymbol{x_{i+1}}$, containing the retrieved ozone vertical distribution at each i-th iteration, is expressed in terms of the Volume Mixing Ratio (VMR), which is more suitable for use with smoothing constraints. The smoothing weights, i.e. square roots of the diagonal elements of $\boldsymbol{\gamma}$, linearly increase with the height above 45 km and remain constant below.

Surface albedo is simultaneously retrieved with ozone using the sun-normalized radiance provided in the L1G data. Two spectral fitting windows at THs around 38 km are employed: 355–365 nm and 455–470 nm, where ozone absorption is weak.

### 3.3   Cloud filter

A cloud filter is applied during the ozone retrieval to reject THs at which a cloud is present in the field of view of the instrument. The applied algorithm is based on the Color Index Ratio (CIR) concept (Eichmann et al., 2016), using OMPS-LP radiance at 754 nm and 997 nm. The Color Index (CI) is defined as the ratio of the radiance at the two chosen wavelengths for the same OMPS-LP spectrum. The CI can be used to detect the presence of scattering particles in the field of view, since we know the expected ratio for a cloud-free atmosphere. First, the CI is calculated at all THs, then the CIR is obtained as:

$$CIR(z_{TH}) = \frac{CI(z_{TH})}{CI(z_{TH} + \Delta z_{TH})} \tag{5}$$

where $\Delta z_{TH}$ is the vertical grid step of 1 km. An example of the results for simulated clouds is reported in Fig. 4: cirrus clouds consisting of hexagonal crystals with an optical depth between 0.01 and 0.15 are taken into consideration. Since the ozone retrieval is run above 12 km, we are generally not interested in liquid water clouds.

The chosen threshold to flag a TH as cloudy is 1.25. This technique was also applied to SCIAMACHY measurements with a different threshold (Eichmann et al., 2016). At the considered wavelengths the measured radiation is related to the scattered light from molecules, aerosol or cloud particles. A question may arise regarding the inability of such an approach to distinguish between high aerosol loads and cirrus clouds. Future investigations will focus on a comparison between the CIR filter and aerosol profiles retrieved as described in the next subsection.

A different approach was used to detect Polar Mesospheric Clouds (PMC). The presence of PMCs can affect limb radiance down to 40 km, causing an interference with ozone retrievals. Therefore, we screen out the PMC contaminated profiles in this study using the PMC detection flag at high latitudes above 50° N and below 50° S where the PMC occurrence is most frequent. PMCs are detected using the radiance profile at 353 nm and several conditions on radiance and its gradients. In absence of PMCs, radiance is expected to decrease monotonically with height above 40 km. As a consequence, the ozone profile is flagged if the radiance between 40 km and 80 km increases with altitude or its gradient increases more then 50% between at least two consecutive layers.

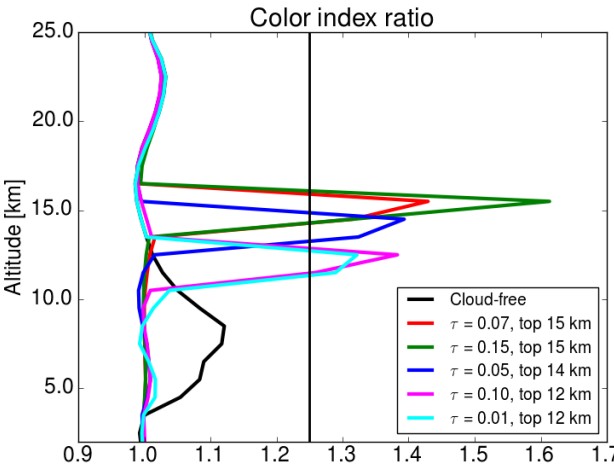

**Figure 4.** Example of Color Index Ratios for different simulations of ice clouds. Top of the cloud and optical depth ($\tau$) ranges are chosen to simulate the impact of thin cirrus clouds in the upper troposphere.

## 3.4 Aerosol treatment

The aerosol extinction coefficient is retrieved employing the general approach as used for SCIAMACHY v1.4 stratospheric aerosol extinction product (Rieger et al., 2017). As a consequence of a coarser spectral resolution, the radiance measured at 750 nm is affected by the $O_2$ absorption band. For this reason the OMPS aerosol extinction coefficient retrieval uses the radiance at 869 nm instead of 750 nm, as it was done for SCIAMACHY and OSIRIS. Stratospheric aerosol extinction is retrieved in the altitude range from 10.5 km to 33.5 km. The measurement at 34.5 km is used as the reference; the effective Lambertian albedo is simultaneously retrieved using the sun-normalized spectrum at 34.5 km. In order to smooth spurious oscillations, the first order Tikhonov regularization is employed. Scattering phase functions are calculated using Mie scattering theory. The particle size distribution is assumed to be lognormal with the median radius ($r_g$) of 0.08 $\mu m$, and distribution width parameter ($\sigma$) of 1.6. The corresponding probability distribution function is given by the following equation:

$$\frac{dn(r)}{dr} = \frac{N}{\sqrt{2\pi}\ln(\sigma)r} \exp\left( \frac{(\ln(r_g) - \ln(r))^2}{2\ln^2(\sigma)} \right) \tag{6}$$

The aerosol particles are assumed to be sulfuric droplets with 0 % relative humidity in the surrounding atmosphere. Below 10 km and above 46 km the aerosol load is set to zero. The refractive indexes are calculated using the Optical Properties of Aerosols and Clouds (OPAC) database (Hess et al., 1998). Before using the retrieved aerosol product, altitudes downwards from the detected cloud top height are rejected and each profile is extrapolated by the scaled a priori. The scaling factor is derived by averaging three altitude levels above the cloud.

Because of the strong forward peak of the aerosol scattering phase function, a correct description of the aerosol scattering is particularly important at high northern latitudes where the scattering angle is small (refer to Fig. 3).

## 4 Results

In this section we present the results of the processing, for the whole year 2016. Version 2.5 of OMPS-LP L1G data has been used, which, in comparison to the previous version, features an improved stray light treatment and pointing corrections as described in Sect. 2.1. Retrievals were performed using data only from the central slit of the instrument because the lateral slits are still considered to suffer from pointing issues.

### 4.1 Retrieval characterization and error analysis

The information content of the measurements as well as the sensitivity of the retrieval can be analyzed using the averaging kernels ($\mathbf{A}$) and the covariance of the retrieval noise ($\mathbf{S_m}$) obtained respectively as (Rodgers, 2000):

$$\mathbf{A} = (\mathbf{K^T S_\epsilon}^{-1}\mathbf{K} + \mathbf{S_r})^{-1}\mathbf{K^T S_\epsilon}^{-1}\mathbf{K} \tag{7}$$

$$\mathbf{S_m} = (\mathbf{K^T S_\epsilon}^{-1}\mathbf{K} + \mathbf{S_r})^{-1}\mathbf{K^T S_\epsilon}^{-1}\mathbf{K}(\mathbf{K^T S_\epsilon}^{-1}\mathbf{K} + \mathbf{S_r})^{-1} \tag{8}$$

The square root values of the diagonal elements of the retrieval noise covariance matrix $\mathbf{S_m}$ will be referred to as the theoretical precision of the retrieval. Following von Clarmann (2014), we do not include smoothing errors in the retrieval error budget. The vertical resolution of the retrieved profile is computed as the inverse of the diagonal elements of the averaging kernel matrix, multiplied by the altitude layer width. Examples of averaging kernels, vertical resolution and theoretical precision are plotted in Fig. 5.

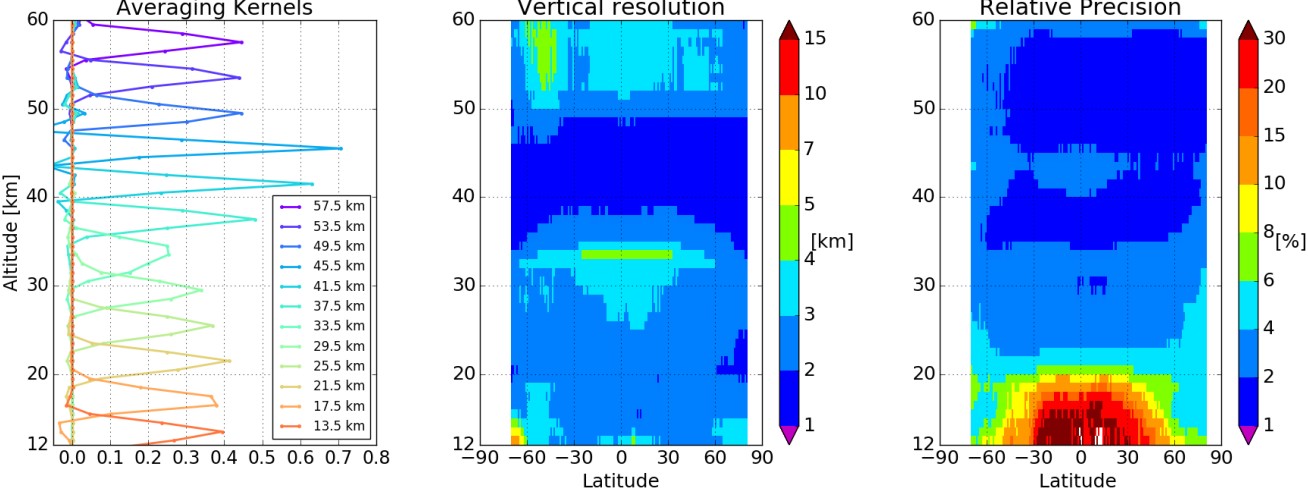

**Figure 5.** From left to right, examples of averaging kernels (plotted every 4 km for sake of clarity), vertical resolution and theoretical precision of the retrieval scheme. AKs are plotted for a measurement at 30° N, whereas vertical resolution and theoretical precision are shown as a function of latitude, i.e. solar zenith angle, for one day (15 September 2016).

The left panel shows AKs for an example profile at 30° N. For the sake of clarity, only each fourth AK is plotted. Middle and right panels show the latitudinal dependence of the vertical resolution and precision, respectively, for one day of OMPS measurements (15 September 2016). Below 30 km the actual vertical resolution of the retrieval scheme is typically about 2–3 km getting worst around 33 km, where the transition between UV and Vis spectral ranges occurs. The best vertical resolution of the profiles is achieved around 45 km, whereas above 50 km it gets coarser, due to the increasing Tikhonov parameter. The theoretical precision of the retrieved ozone profiles doesn't show any significant dependence on the solar zenith angle (or latitude) above 25 km. It lies in the range of 1–4 % up to 60 km and tends to increase at lower altitudes, particularly in the tropical Upper Troposphere - Lower Stratosphere (UTLS) region: at these levels, the ozone concentration drops significantly and the retrieval precision gets lower, with relative errors up to 10–30 %. This purely random uncertainty is expected to be significantly reduced when averaging several profiles, as it is done in the validation section of this paper. For example, considering 10000 profiles and a relative precision of 30 % for each single profile, the random uncertainty on the averaged profile is equal to 0.3 %. Therefore, the random noise error is rather negligible when analyzing the validation results.

## 4.2 Comparison with NASA OMPS-LP ozone product

To retrieve ozone profiles from OMPS-LP observations, the NASA team implemented the Environmental Data Record algorithm, based on the Optimal Estimation approach with a priori constraints. In this procedure, a series of secondary parameters such as surface albedo, cloud height and TH correction are derived before the main retrieval of ozone profiles (Rault and Loughman, 2013). Two spectral ranges are used for the latter task: UV wavelengths between 29.5 and 52.5 km and wavelengths in the Chappuis band between 12.5 and 37.5 km. The normalization of the radiance is performed with respect to high altitude TH measurements: 55.5 km in UV and 40.5 km in Vis. The measurement vector is obtained using the doublet and triplet method respectively for the Hartley-Huggins and Chappuis bands; more details are given in Table 2. An additional TH correction is applied by NASA team on L1G data, with values that follow an approximate linear decrease along the orbit, as described in the Release Notes of Level 2 (L2) data (DeLand et al., 2017). The quality flag related to the South Atlantic Anomaly is taken into consideration for the following comparison (Kahn and Kowitt, 2015).

**Table 2.** Wavelengths used in the NASA-OMPS ozone retrieval, according to DeLand et al. (2017)

| Parameters | Values |
| --- | --- |
| Doublet $\lambda_0$ | 353 nm |
| Triplet $\lambda_l$ | 510 nm |
| Triplet $\lambda_r$ | 675 nm |
| Wavelength used in UV (nm) | 302, 312, 322 |
| Wavelength used in Vis (nm) | 600 |

In version 2.5 of NASA L2 data, independent profiles for the Vis and UV retrievals are provided. Figure 6 shows a comparison between NASA-OMPS retrievals and our results (in the following called IUP-OMPS), considering the two retrieved profiles independently. Panel (a) presents an example of averaged profiles in terms of the number density for the tropics. In panels (b) and (c), relative differences are shown for the tropical region, southern and northern mid- and high-latitude bands. Throughout the paper, relative differences are computed as:

$$\text{Rel diff} = \frac{2 * (\text{IUP-OMPS - Reference data set})}{(\text{IUP-OMPS + Reference data set})} * 100 \tag{9}$$

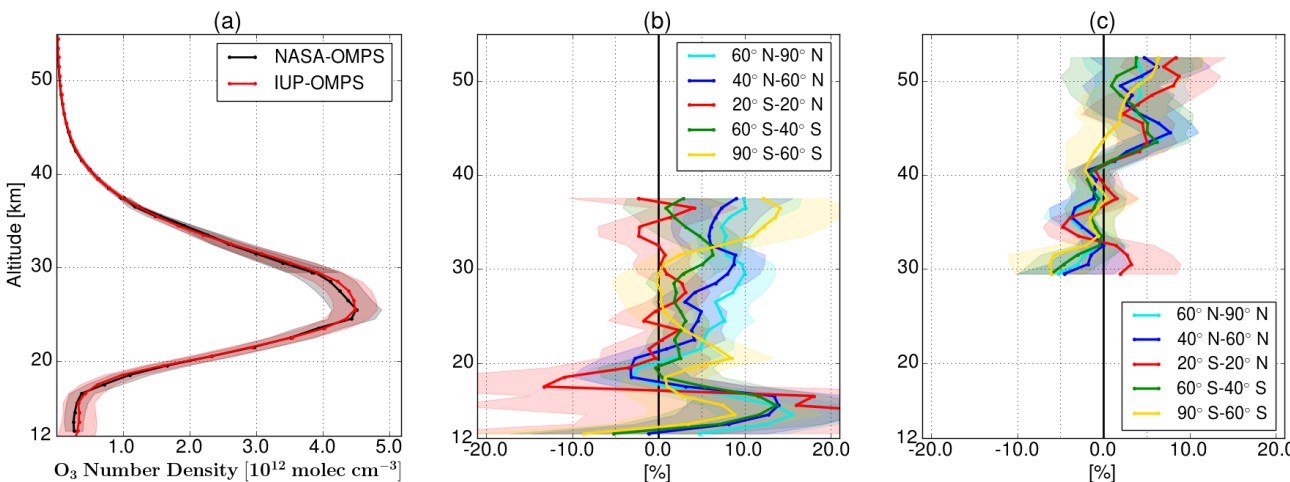

**Figure 6.** Panel (a): IUP-OMPS and NASA-OMPS retrieved number density profiles averaged in the tropical region. Panel (b) and (c): the relative differences (Eq. 9) for the Vis and UV retrievals, respectively, are shown in five latitudinal bands (60° N–90° N, 40° N–60° N, 20° S–20° N, 60° S–40° S and 90° S–60° S), with corresponding standard deviations as shaded areas.

Considering the discrepancies with the NASA Vis retrieval, in panel (b), an excellent agreement within 3 % is achieved in the tropical region above 20 km. At northern mid-latitudes the agreement is slightly worse with differences up to 5 % between 18 and 27 km and 5–9 % above 28 km. This positive bias slightly increases towards northern polar regions. The differences at southern mid-latitudes show a similar altitude behavior as those in the northern hemisphere but have a smaller magnitude. Towards the South Pole, we notice the worst agreement above 30 km. These discrepancies are possibly related to the merging of the spectral information from UV and Vis ranges at these altitudes in IUP retrievals. In the UTLS region we notice larger differences between the two profiles especially in the tropics: at these altitudes the ozone concentration and, thus, the retrieval accuracy gets lower. As a consequence, specific settings of the two retrievals such as spectral ranges, a priori values, aerosol and cloud retrievals play a larger role and are the most probable reason for the observed disagreement. Unfortunately, it is not possible to identify and relate each discrepancy at different altitudes to specific settings of the 2 algorithms: a stepwise adjustment of the settings is not always feasible, because the intermediate retrieval versions would result in oscillating or non-converging solutions.

Considering the NASA UV retrieval, the differences shown in panel (c) are very similar for all latitude bands and an agreement within ±5 % is observed at most altitudes, except around 50 km in the tropics and at 45 km at northern mid- and high-latitudes, where the relative differences increase to 5–8 %. This may be related to the different usage of UV spectral ranges or TH normalization.

The jump between Vis and UV retrievals, evident when comparing panels (b) and (c), especially at northern mid-latitudes, is also reported by the NASA team in the corresponding Release Notes (DeLand et al., 2017): a preliminary comparison of their results with MLS assesses that values retrieved in the Vis range in the overlapping region (29.5–37.5 km) are systematically lower at mid- and high-latitudes.

## 4.3   Comparison with MLS

The MLS instrument was launched on board the Aura satellite in July 2004 to observe the thermal emission from atmospheric trace gases in the millimeter/sub-millimeter spectral range. It scans the Earth limb 240 times per orbit, providing retrievals of day- and nighttime profiles of several gases. For a detailed description of the MLS instrument refer to Waters et al. (2006). In this paper, the version 4.2 of MLS L2 data is used for the validation. Quality flags and recommendations reported in Livesey et al. (2017) are taken into consideration for the following analysis. Because of the large amount of available data, tight criteria

are applied to collocate measurements. The geographic distance between the centers of the two instrument footprints is limited to be within 1° latitude and longitude and the time difference is required to be within 6 h. In addition, the difference in the potential vorticity at 20.5 km is required to be less than 5 PVU, in order to avoid collocation of measurements inside and outside the polar vortex. Information about potential vorticity is taken form the European Center Medium Weather Forecast (ECMWF) database (ERA interim). OMPS-LP states affected by the presence of PMCs and observations at altitudes flagged

as cloudy are rejected. In case of multiple MLS collocations for the same OMPS-LP measurement, only the closest one is taken into consideration. To be consistent with NASA and sonde comparisons, MLS profiles are converted from VMR vs. pressure into number density vs. altitude (using MLS geopotential height), interpolated at the regular altitude grid of IUP-OMPS retrieved profiles and finally zonally averaged. Five latitudinal bands are selected for the comparison: 60° N–90° N, 40° N–60° N, 20° S–20° N, 60° S–40° S and 90° S–60° S. Fig. 7 shows the averaged profiles for tropics and northern mid-

latitudes and the relative differences (Eq. 9) in the five latitudinal bands. Standard deviations are reported in the plots as shaded areas. The number of collocations per band is ∼ 10000. The zonally averaged relative differences of IUP-OMPS with MLS are found to be generally within 5% between 20 and 58 km for all latitude bands. In the tropics a fairly constant positive bias of 2–4% is observed at all altitudes above 28 km. At northern mid-latitudes we notice a negative discrepancy of about 6–7 % around 28–30 km, which becomes more evident towards polar regions; up to 58 km the relative difference exceeds 3 % only at

45 km. At southern mid-latitudes IUP-OMPS shows about 5 % higher ozone number density around 25 km, whereas at other altitudes between 20 and 58 km an agreement within 3 % is achieved. Below 20 km, the agreement with MLS gets worse, with relative differences above 20% in the tropics, even though the absolute difference is rather small (see panel a).

Fig. 8 shows the relative differences between IUP-OMPS and MLS zonal means in 2.5° latitude bins as a function of altitude. Three time periods are considered in the panels. In panel (a), showing the whole year 2016, we can see that between

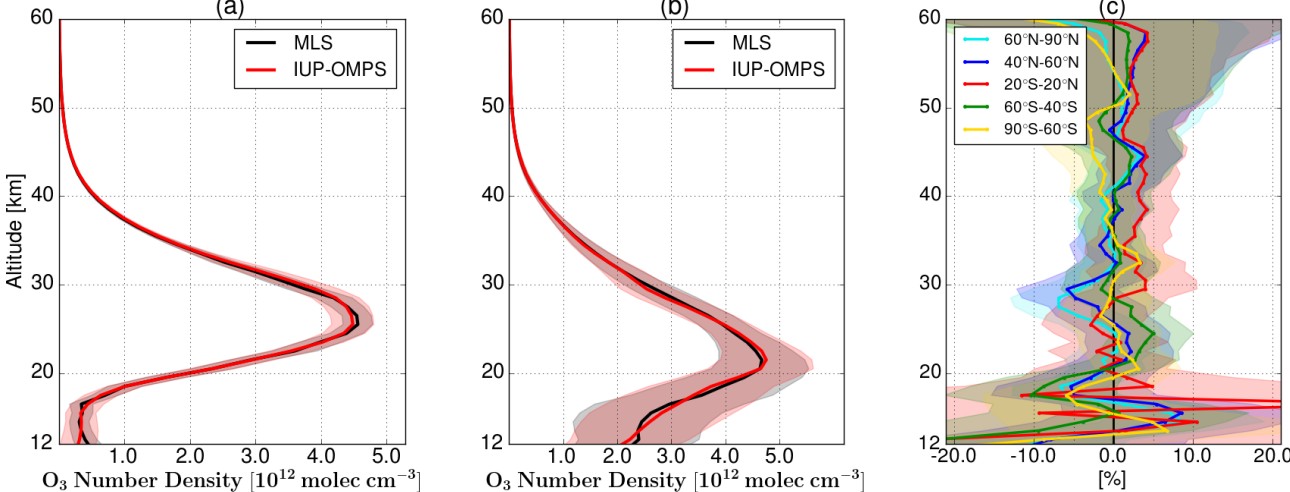

**Figure 7.** Panel (a) and (b): collocated IUP-OMPS retrieved profiles and MLS ozone product in the tropical region and at northern mid-latitudes, respectively. Panel (c): relative difference profiles (Eq. 9) in five latitudinal bands ($60°$ N–$90°$ N, $40°$ N–$60°$ N, $20°$ S–$20°$ N, $60°$ S–$40°$ S and $90°$ S–$60°$ S), with standard deviations shown as shaded areas.

20 and 60 km the differences are mostly within $\pm 5$ % and never exceed 10 % at all latitudes. Starting the discussion of the discrepancies from the bottom of the plots, oscillating differences larger than 30 % are found in the tropical UTLS region. This large discrepancy can be related to several factors such as a high dynamic variability of ozone, generally low sensitivity in the lowermost retrieval altitude range or issues with the cloud filtering. As already mentioned, in this region the ozone concentration gets very low and the accuracy of the retrieved profiles degrades. Around 28–33 km a dip in IUP-OMPS ozone values is visible towards the northern high-latitudes, especially during winter months (panel c), whereas higher values are found in the tropics: this altitude range corresponds to the overlap region between the contributions from UV and Vis spectral windows and their merging can lead to some inconsistencies. In addition, an non-optimal albedo retrieval could cause biases during winter at northern polar latitudes. Around 45 km, higher values are shown by IUP-OMPS in tropics and at northern mid-latitudes. This issue, already found in Fig. 6 panel (c), can be related to a problem in the junction between the spectral ranges in the Hartley and Huggins bands, that occurs at 46 km. In panel (c), we also notice a discrepancy in the equatorial region and towards the North Pole above 50 km; this disagreement can be partly related to the stray light affecting the TH used for the normalization.

To summarize, the presented comparison shows a general validity of IUP-OMPS retrieval between 20 and 58 km in the tropics and down to 15 km at mid-latitudes, even if during different seasons the relative bias with respect to MLS may exceed 10 % in some atmospheric regions.

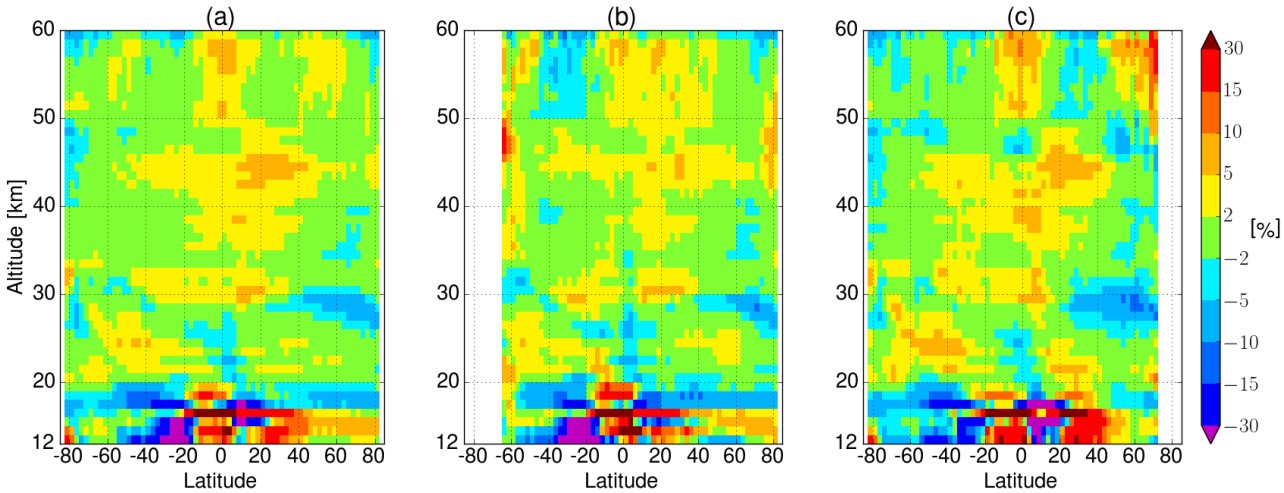

**Figure 8.** Relative differences (Eq. 9) averaged over 2.5° latitude bins, plotted as a function of altitude. Panel (a) whole year, panel (b) June, July and August, panel (c) January, February and December.

### 4.4 Comparison with ozonesondes

In order to provide a more reliable validation of our product at altitudes below 30 km, we are taking into consideration ozonesonde measurements. Ozonesonde data are obtained from WOUDC (World Ozone and Ultraviolet Radiation Data Center) and SHADOZ (Southern Hemisphere ADditional OZonesondes, Thompson et al., 2007) archives. We selected looser
5  collocation criteria compared to MLS, because of the sparseness of ozonesonde measurements. Therefore, OMPS-LP measurements are required to be within 5° in latitude and 10° in longitude from the ozonesonde station and within $\pm 12$ h time span around the sonde launch. For each sonde profile, all collocated OMPS-LP observations are averaged before the comparison. In order to account for the different vertical resolution of the compared profiles, ozonesonde measurements are convolved with the AKs of the IUP-OMPS retrieval scheme as follows. First, we calculate the linear interpolation matrix $\mathbf{L}$ to map the low
10 resolution OMPS profile onto the fine sonde grid. Then this matrix is inverted using the pseudo-inverse formulation (Rodgers, 2000), obtaining $\mathbf{L}^*$ as:

$$\mathbf{L}^* = (\mathbf{L}^T \mathbf{L})^{-1} \mathbf{L}^T \tag{10}$$

The ozonesonde high resolution profile $x_{fine}$ is then convolved as follows:

$$x_{coarse} = \mathbf{A} \, \mathbf{L}^* \, x_{fine} \tag{11}$$

15 The upper altitude of the convolved profile is chosen at the OMPS-LP grid level whose corresponding AK altitude range is fully covered by the sonde profile. An approach alternative to the AK convolution assumes a simple vertical average, considering 2.5 km (i.e $\pm$ 1.25 km) ranges around each grid point (value corresponding to an average vertical resolution of the retrieval

scheme below 30 km, refer to Fig. 5). The altitude where a cloud is detected and all altitudes below are screened out. Latitude bins are selected in the same manner as in the previous comparisons.

Fig. 9 shows averaged collocated profiles in the tropical and northern mid-latitude bands with corresponding standard deviations. On the left side of these plots, the number of available collocations at each altitude is reported, which is about 220 and 370 for tropical and northern mid-latitude bands, respectively. Overall, 37 ozonesonde stations were considered, corresponding to over 1300 single collocated profiles.

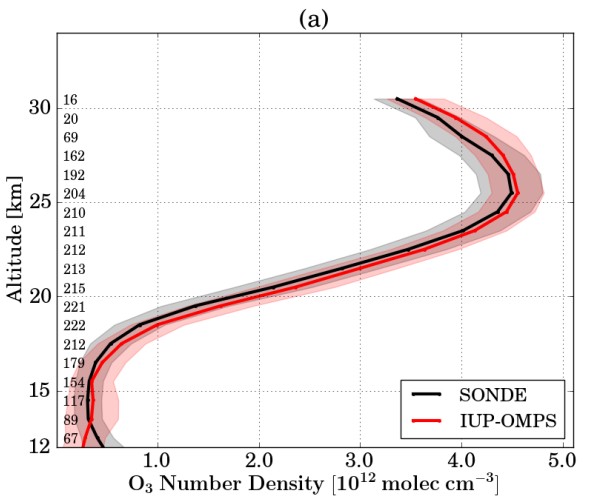
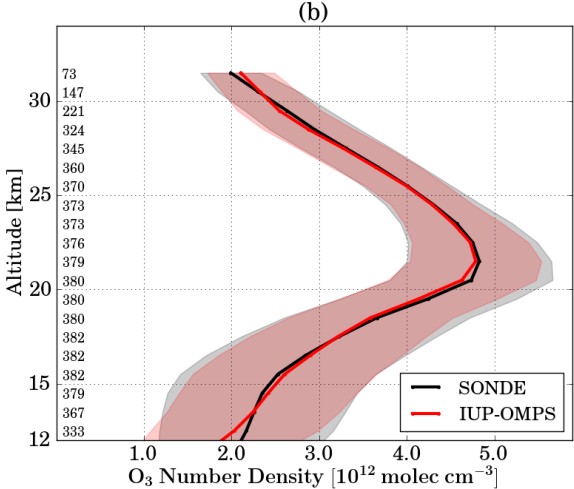

**Figure 9.** Comparison between collocated IUP-OMPS profiles and ozonesonde measurements in the latitudinal bands 20° S–20° N in panel (a) and 40° N–60° N in panel (b); standard deviations are shown as shaded areas.

Fig. 10 shows the relative differences (Eq. 9) in five latitudinal bands, in panel (a) using the averaging kernel convolution approach and in panel (b) the vertical averaging. Differences between the two panels of this figure show that the averaging procedure can be critical in the comparison between 15 and 20 km, where the gradient in the ozone profile is usually strong. As shown in Figs. 9 and 10, an excellent agreement is found at northern mid-latitudes, with relative differences mostly within ± 3 % between 13 and 30 km. Towards northern polar regions, a similar agreement is found, with a positive bias of 3 % down to 12 km. At southern mid-latitudes, we notice a fairly constant positive difference between 20 and 30 km, with values of 4–6%. A similar positive bias at southern mid-latitudes is also visible in Fig. 7. At southern polar latitudes the agreement gets slightly worse, with a discrepancy up to 7 % above 13 km. Focusing on the tropical region, a bias between the two data sets is clearly visible, with differences around 2–20 % above 13 km. The positive bias above 17 km is unexpected considering the good agreement found when comparing to MLS data in the same region, even though a positive anomaly is also visible in Fig. 8 about 18–20 km between 20° S and 0° S.

Looking at panel (b) of Fig. 10, the same patterns are depicted but stronger oscillations below 20 km are found, due to the smaller vertical range over which the sonde profiles are averaged.

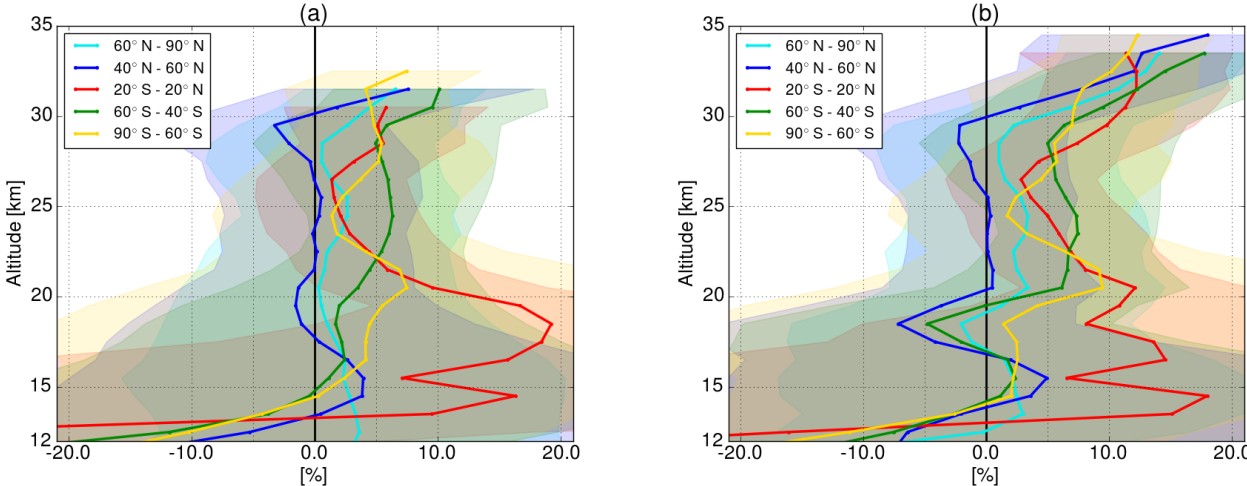

**Figure 10.** Relative differences between collocated IUP-OMPS profiles and ozonesonde measurements in five latitudinal bands (60° N–90° N, 40° N–60° N, 20° S–20° N, 60° S–40° S and 90° S–60° S), using in panel (a) averaging kernel convolution and in panel (b) vertical averaging. Corresponding standard deviations are shown as shaded areas.

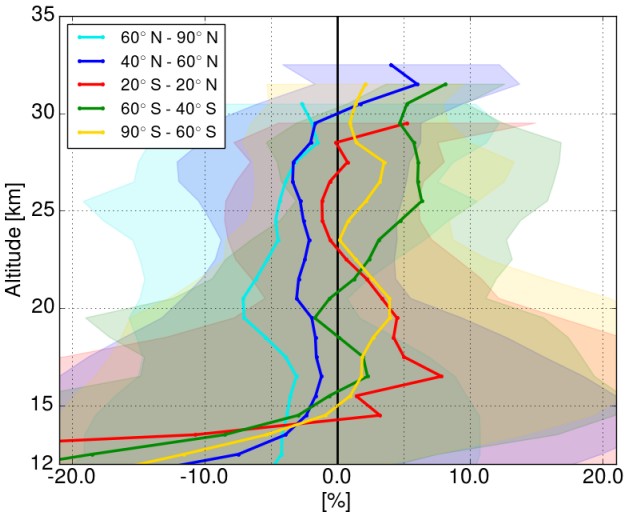

**Figure 11.** Relative differences between collocated IUP-OMPS profiles and ozonesonde measurements in five latitudinal bands (60° N–90° N, 40° N–60° N, 20° S–20° N, 60° S–40° S and 90° S–60° S) for the 2013 data set, with corresponding standard deviations as shaded areas.

With respect to the bias found in the tropical region, the processing of the OMPS-LP 2013 data set is also performed using the same retrieval settings. The analysis of these results and their validation against ozonesondes reveal a much smaller bias in

the tropics. Relative differences between IUP-OMPS and sonde profiles in the same five latitudinal bands are shown in Fig. 11, following the averaging kernel convolution approach. Since most of the tropical sondes considered over the year 2016 come from the SHADOZ archive, we also take only measurements from the same archive for the 2013 validation: over the whole year, around 200 collocations are available from 10 stations in the tropics and around 1000 collocations from 30 stations at mid and high-latitudes. In Fig. 11, focusing the attention on the differences in the tropical region, we can see that between 14 and 30 km the bias is mostly within 5 %; although a discrepancy of about 8 % is still evident at about 16 km. Considering all the latitude bands, an agreement within 8% is seen at all altitudes above 14 km, as found in Fig. 10, panel (a). A comparison with MLS shows a very similar pattern to the one observed for the 2016 data set. A possible reason for this difference between the data from 2013 and 2016, might be a jump of about 100 m in the pointing of the instrument that the NASA team detected in September 2014 and that was not corrected in L1 data. The effect of this small jump would be particularly evident at altitudes where ozone profile shows the strongest gradient, that is around 18–22 km in the tropics, and it is in agreement with the slight shift of the two profiles visible in panel (a) of Fig. 9, even though this is not found in the comparison with MLS. Concluding, we find a general consistency of IUP-OMPS retrieval results with ozonesonde measurements in all considered latitude bands, except for the 12–20 km altitude range in the tropics, where the agreement with SHADOZ ozonesondes is ambiguous.

## 5 Conclusions

The retrieval algorithm originally developed at the University of Bremen to obtain vertical distributions of ozone from SCIA-MACHY limb measurements was tailored and applied to OMPS-LP observations. The v2.5 L1G data set from the whole year 2016 was processed, analyzed, validated and the results were presented here. Ozone profiles were retrieved between 12 and 60 km, considering only the central slit of the instrument and observations at a solar zenith angle less than 80°. A comparison with NASA v2.5 L2 official product was carried out: we found an overall good agreement with the UV product at all latitude bands, with discrepancies typically within ± 5 %, except around 45 km and 50 km. The comparison with the Vis product above 20 km showed generally good consistency, even though a discrepancy of 7–12 % was observed above 27 km at northern mid-latitudes and polar regions. We presented the results of the validation against MLS v4.2 ozone profiles and ozonesonde measurements from SHADOZ and WOUDC archives. A good agreement was found with the MLS ozone product: relative differences were generally within ± 5 % between 20 and 58 km. On the other hand, we observed a larger discrepancy between IUP-OMPS retrievals and MLS in the tropical UTLS region, related most probably to smaller ozone amounts, larger dynamical variations and the decreasing sensitivity of limb retrievals from both instruments in this region. In regard to the comparison with ozonesondes, we found at northern mid- and high-latitudes differences within ± 5 % between 13 and 30 km and at southern mid- and high-latitudes a positive bias about 5–7 % for the same range. Focusing on the tropical region, a significant positive bias with SHADOZ measurements was detected, unexpected after the good agreement observed with MLS data. However, the processing and validation of the 2013 data set, using the same retrieval settings, revealed a much better consistency. The reasons for this behavior are still under investigation, but are possibly related to a jump in the pointing of the instrument occurred in 2014. In light of the results presented here, an additional work for tuning of some retrieval settings is

needed before processing the whole data set and attempting the merging with the SCIAMACHY time series. Since the same 1-D retrieval approach has been used for both data sets, we expect this to ease the merging. Unfortunately, only a couple of overlapping months between the two instruments are available, so that a third product must be used for the merging. After the good agreement found in the comparison of our retrievals with MLS, we are considering the use of the latter instrument as a transfer function to handle calibration issues in the merging procedure.

## 6 Data availability

Ancillary information and v2.5 L1G OMPS-LP data were downloaded from https://ozoneaq.gsfc.nasa.gov/data/omps/, where L2 data are also available.

For the validation sections, MLS L2 data were taken from https://disc.gsfc.nasa.gov/datasets. WOUDC data were retrieved on Feb 5, 2018, from http://woudc.org; a list of all contributors is available on the following website: doi:10.14287/10000001. SHADOZ were retrieved on Feb 5, 2018 from https://tropo.gsfc.nasa.gov/shadoz/Archive.html. Our results are available upon request from the University of Bremen.

*Author contributions.* CA adapted the retrieval algorithm to OMPS-LP observations, processed the data set, performed the validation of the results and wrote the manuscript. AR provided the retrieval algorithm exploited in this study, supervised and guided the retrieval process and reviewed the paper. EM provided retrieved aerosol extinction profiles. K-UE contributed with the algorithm for cloud filtering that was adapted to OMPS-LP observations. TvC contributed to the discussion of the regularization matrices for the retrieval scheme and the proper use of averaging kernels to smooth the ozonesonde profiles and reviewed the paper. JPB, who proposed the research and leads the project, analyzed the results and contributed to the writing of the manuscript and the scientific outcomes.

*Competing interests.* The authors declare that they have no conflict of interests. TvC is associated editor of AMT but is not involved in the reviewing of this particular paper.

*Acknowledgements.* This work was partially funded by ESA within the Ozone CCI project and was supported by the University and State of Bremen. We would like to acknowledge NASA OMPS SIPS team (in particular G. Jaross, N. Kramarova and P.K. Bhartia) for the support provided during the data analysis as well as for the concession of the new v2.5 of OMPS-LP L1G data before its official release. Part of the data processing has been done on the German HLRN (High Performance Computer Center North). The GALAHAD Fortran Library was employed in the retrieval scheme.

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
