# Peer review of "Retrieval of ozone profiles from OMPS limb scattering observations"

_Atmospheric Measurement Techniques, 2017_

## Referee Comment (RC1) · Anonymous Referee #1 · 13 Sep 2017

This article presents an Ozone Profile Retrieval Algorithm using atmospheric limb scattering of solar UV and visible radiances measured by OMPS Limb Profiler (LP), to produce high vertical resolution ozone profiles in the stratosphere and upper troposphere. This algorithm is based on a Tikhonov regularization, which was originally developed for the SCIAMARCHY Limb instrument. Therefore, the application of the similar retrieval scheme to these measurements will offer a great opportunity of long-term trend analysis if the consistency of two dataset could be well demonstrated with the cross-calibration using the overlap period between OMPS and SCIAMARCH. In this study, ozone profiles are retrieved in the unit of VMR at altitude-based vertical levels from surface to 60 km from OMPS-LP L1B v2.5 data for seven months from July 2016 and January 2017. The vertical resolution of retrievals vary from ~ 2.5 km at lower altitude levels (< ~ 30 km) and ~ 1.5 km to upper altitude levels (from 40 km to just below top levels). The theoretical retrieval precisions are estimated to be 1-5 % above 25 km, but rapidly increase to 15 % at 20 km. In order to evaluate the developed algorithm seven months of retrieved ozone profiles are compared to OMPS LP Version 2.0 daily ozone product, MLS v4.2 stratospheric ozone profile standard product and global ozonesonde measurements.

**General Comment**

The overall parts of the paper are written unclearly and illogically. For example, vertical grids where OMPS ozone profiles are retrieved and the unit of ozone should be introduced in the algorithm description section, at once. However, I need to search from.

   -In line8 on page 9, the vertical range between 12 and 60 km

   -In line3 on page 10, unit of ozone: **VMR**

   -In line13 on page 14, authors described "MLS are converted from VMR vs. pressure into **number density** vs. altitude, interpolated at the regular altitude grid of OMPS", in addition, the regular altitude grid is not mentioned before.

   - In line 10 on line 16, 2.6 km corresponding to an average vertical resolution of the retrieval scheme.

   This article should be checked line-by-line to become more scientific. For example, in abstract, authors mentioned "ozone in the 12-60 km can be retrieved due to using spectral window in the Hartley, Huggins and Chapais ozone absorption band" In the view of the spectral window, this instrument is optimized to detect ozone over the troposphere including surface rather than the stratosphere. Limb measurements has lack sensitivity to troposphere due to its viewing geometry. In addition, authors described the OMPS-LP official algorithm as an inversion scheme with a priori constraints and a Tikhonov regularization, but in OMPS documentation, it is based on an optimal estimation based regulated by a set of a-priori constraints. These two schemes are not same. Please change "Level 1" to "Level 1b" because these two product are not same.

Insufficient analyses on retrieval/validation were performed, which is commented in the main comment section. I found the text not be precise enough concerning unformatted types, grammatical error, English usage, which is commented in the minor comment section.

**The following is the main suggestions for improvements.**

**0. Abstract**

- Remove "this algorithm was originally developed ~~ to produce a combined data set" in the abstract part and add more about the retrieval related description or results. For example, the vertical resolution of retrievals vary from ~ 2.5 km at lower altitude levels (< ~ 30 km) and ~ 1.5 km to upper altitude levels (from 40 km to just below top levels). the theoretical retrieval precisions are estimated to be 1-5 % above 25 km, but rapidly increase to 15 % at 20 km.

- "The optimization of the retrieval algorithm ~~. ➔ This algorithm use altitude-normalized radiances in the UV and VIS wavelength range.

- indicating a good agreement ➔ specify the altitude range showing a good agreement e.g.) a demonstrating a good agreement from 15 km to 58 km.

- did not mention about the comparison with OMPS/NASA product.

**1. Introduction**

- Authors mentioned that the main objective of the study is to create the long-term dataset using OMPS and SCIAMARCH. To do this, how to overcome the discrepancy of two instrument calibration? It is very difficult because of little overlapping period between OMPS and SCIAMARCH. Please add shortly how to overcome the discrepancy of two instrument calibration.

- Authors too much simplified the summery of the previous studies related to your data product, compared to the history and importance of ozone chemistry. It might be better to remove the ozone chemistry-related party (this part is unclearly written) and to focus on 1) history of satellite ozone observation using limb instrument, 2) why we need limb instrument compared to nadir instrument for ozone observation  3) why we need solar scattered limb measurements compared to infrared/microwave emission limb measurements for ozone observation, 4) history of SCIAMARCHY limb ozone profile product; algorithm development/ validation, the long-term stability of both instrument and ozone dataset, 5) OMPS LP ozone profile product from OMPS science team at least and others if possible (e.g. Daniel et al. 2017 recently submitted to AMT), 6) the effort of this study to optimize the SCIAMARCHY algorithm for OMPS.

- Line 33-35, page 2: the limb combines the advantage of the other two techniques ~~ with relatively high vertical resolution and horizontal coverage; reader who have no idea about satellite instrument could be confused that which instrument has higher vertical (horizontal coverage) resolution compared to Limb.

**2. OMPS LP instrument**

**2.1 General features.**

- Line 25 (page 3) "The main objective of the mission is to monitor the ozone vertical distribution within the Earth middle atmosphere at high accuracy level" ➔ it is not true because the mission mentioned belongs just to the SNPP.

- Move line 23-27 to introduction and focus just on OMPS LP.

- line 1, page 5: the spectral range between 280 nm and 1000 nm ➔ the spectral range of 290 nm to 1000 nm.

- line 5-8, page 5: The use of such a technology (observation at the same time without vertical scanning and CCD) pose a great challenge as regards the SNR; indeed, scattered solar radiance from the Earth limb decreases by at least five orders of magnitude along the considered vertical range, due to the decrease of atmospheric density. ➔ It is illogically written, about the cause-and-effect.

**2.2 Calibration and main issue.**

- This party should be simplified or removed and then move some parts in other sections. Example, 1) In algorithm description, we can delivery some calibration issues related to the treatment of this algorithm to overcome these issues 2) In lines 8-9 on page 15, authors mentioned the disagreement between OMPS and MLS can be partly related to pointing issues, due to the solar heating of the instrument at high latitudes or stray light in section 4.2. In this paragraph, this paper can provide more detailed calibration issues related to this discrepancy.

- Line 2 on page 6: Delete "Level 1B data are provided by NASA team" because the data is publically available.

- Line 24 on page 7: delete "In the preparing time of this paper the new data version was not fully released and only seven consecutive months were available." This kind of sentence is not suitable in the scientific article. Move or re-mention "Retrievals were performed using data from the central slit of the instrument only because the lateral slits can still suffer from pointing issues" in the algorithm description or in the beginning of 4. Results.

**2.3 OMPS-LP geometry of observations**.

- line 31 on page 7: Azimuth angles could be defined separately as solar azimuth angle and satellite azimuth angle.

- line 34 on page 7: positive angles are East of the north, so that values are inside the -180 to 180 range ➔ it is hard to understand this sentence.

- Why this paper need this section? The information given in this part is never mentioned in other sections.

**3. Retrieval method**

**3.1 The retrieval algorithm**

- Describe the theoretical inversion scheme first including from line 25 on page 9 to line 18 on page 10, generally and then describe how this algorithm prepare the measurement vector, measurement error vector, forward model vector, and state vector, it might be better to describe them in separated two sections.

- Move the retrieval characterization and error analysis including Figure 6 in section 4.1 with the changed section title from 4. Satellite data set comparison to 4. Results; 4.1. Retrieval Characterization and Error Analysis 4.2 Comparison with OMPS-LP Ozone Product 4.3 Comparison with MLS 4.4 Comparison with Ozonesonde. This study described that "The information content of the measurements as well as the sensitivity of the retrieval can be analyzed using ~~ and the covariance of retrieval noise". It is true for AK, but not true for retrieval error. Sm is generally called "solution error covariance" including random-noise retrieval error covariance and smoothing error covariance. It should be detailed in the paper and an example should be presented in the right panel of Figure 6. It is useful to add the retrieval characterization and error analysis for mid/high latitudes due to the dependence of the sensitivity of solar measurements on solar zenith angles.

- The DFS and solution errors of OMPS LP seems to be much better than OMI UV nadir viewing sensors in the troposphere (Liu et al., 2010). If it is true, we should use OMPS LP measurements for tropospheric ozone retrievals, but it is know that the limb measurements has lack sensitivity to lower troposphere, due to its viewing geometry. I think that the DFS and Retrieval errors are over/under estimated.

- The definition of normalized radiance is unclear ➔ Measurement vector is defined as the logarithm of the altitude-normalized radiances to an upper TH for canceling calibration errors and reducing the effect of surface/cloud reflectance.  Table 1 summaries ~~.  In this paragraph, this paper should mention that this algorithm rejects the wavelength between 580 and 670 nm and between 620 and 630.0 to remove the effect of water vapor and O2 absorption when you describe which wavelengths are implemented in this algorithm.

- Describe that ozone profiles are retrieved at which vertical grids; the number of levels, the vertical intervals, the unit of the grid in the same paragraph.

- Authors described that ozone retrievals are retrieved from 12 and 60 km in the all sections, but analyzed the retrievals from surface and 60 km.

- Line 20-24, page 9: "A shift and squeeze correction is applied in the Chappuis band to the modeded spectrum with respect the measured one: this pre-processing is performed for each observation at each TH independently" ➔ a. describe why the wavelength calibration is implemented just for VIS wavelengths.  b. Probably the modele spectrum is high resolution solar reference data?

- line 23-25, page 10: ➔ surface albedo is simultaneously retrieved with ozone using two spectral fitting windows (~~) where  ozone absorption is weak.

**4. Satellite data set comparison**

**4.1 NASA retrieval and comparison**

- Line 15: " At the moment of the submission of the paper, only version2 of Level 2 (L2) NASA product was available, so a comparison with the most recent retrieval could not be performed"

This description is not suitable. This study should use the version 2.5 or should confirm from OMPS science team that there is insignificant difference between v2.0 and v2.5 product. This paper mentioned that OMPS/NASA algorithm is based on an inversion scheme with a prior constraints and a Tikhonov regularization, which should be changed to "an optimal estimation based regulated by a set of a-priori constraints".

-Based on Figure 8, there are significant differences between OMPS/NASA and OMPS/IUP products, which different implementations between algorithms causes these differences? Based on Figure 9, it seems that MLS shows better agreement with OMPS/IUP in the stratosphere (ozone peak layer) and with OMPS/IUP in troposphere. Both OMPS and MLS has lack sensitivity to lower troposphere so the retrievals determine mostly from a priori information, the similarity between two product might come from the similarity of a priori data between two algorithms.

- OMPS/NASA should be compared with MLS and ozonesonde to see which one provides better retrieval qualities

**4.2 MLS comparison**

- change the reference of Waters et al. (2006) to MLS v.2 data quality and description documentation. This doc specifies how to use MLS product as following. This study use this data screening method?

**Data screening**

**Pressure range:** 261 – 0.02 hPa.

> Values outside this range are not recommended for scientific use.

**Estimated precision:** Only use values for which the estimated precision is a positive number.

> Values where the *a priori* information has a strong influence are flagged with negative precision, and should not be used in scientific analyses (see Section 1.5).

**Status flag:** Only use profiles for which the **Status** field is an even number.

> Odd values of Status indicate that the profile should not be used in scientific studies. See Section 1.6 for more information on the interpretation of the Status field.

**Quality:** Only profiles whose **Quality** field is greater than 1.0 should be used.

**Convergence:** Only profiles whose **Convergence** field is less than 1.03 should be used.

**Clouds:** Scattering from thick clouds can lead to more systematic effects in the UTLS.

> Most of the affected profiles are removed by the Quality and Convergence screening recommendations (although Convergence issues occur only rarely).

> One should *reject* profiles with odd Status *or* even Status profiles with Convergence above the convergence threshold *or* Quality below the quality threshold. Conversely, one should *keep* profile values with even status *and* good Convergence *and* good Quality. These criteria typically remove 1 to 2 % of global daily data, with tropical latitudes showing somewhat larger data removal fractions of about 5%. This screening generally maintains sufficient coverage for a near-complete daily map (for any given day), even in the UTLS.

> Compared to data screening recommendations for past data versions, the screening of v4.2x data generally removes somewhat fewer ozone profiles on a typical day.

- In this section, we firstly give a description of the vertical grid and the unit of ozone profile used in comparison, but this part should be moved before comparison with OMPS/NASA. I think that this paper create one section to describe the comparison methodology.

- This paper mentioned "an increase of the smoothing parameter is expected to partially attenuate the latter problem", about the large difference between OMPS and MLS profiles around 50 km. This explanation is so vague. Smoothing parameter indicates smoothing errors?

- Figure 10 could be re-analyzed for several months (July and Dec or summer and winter) due to sufficient collocation.

- This paper can mention about the validity of OMPS retrievals above ~ 15 km and below 58 km based on comparison with MLS.

- Line 4 page 14: What is the modified potential vorticity?

- Line 9 page 15: "not screened polar mesospheric clouds" ➔ based on the cases provided in this paper, it is hard to relate the large difference between OMPS and MLS to polar mesospheric clouds (PMC). That is because the presence of PMC is limited to polar summer season, but your analysis is performed for all seasons. This article did not mention that why the presence of PMC is important for OMPS retrievals and why MLS could be not impacted by PMC, maybe need some reference.

**5. Ozonesonde comparison**

   - Convolution process of higher resolution profiles with averaging kernels could be described after equation (4).

   - This paper mentioned Figure 12 (a) as "averaging kernel smoothing and (b) as "vertical averaging". Please correct this way to "Comparison of OMPS ozone profiles with ozonesonde smoothed with OMPS averaging kernel and (b) without smoothing, respectively".

   - This paper can add about insignificant impact of the smoothing of ozonesonde profiles to OMPS vertical resolution on the comparison results in the stratosphere due to the comparable vertical resolution of OMPS LP ozone profile retrievals to ozonesonde, compared to the comparison between nadir UV ozone product and ozonesonde. This fact can emphasize the importance of limb instrument on the stratospheric ozone observation.

   - Should summarize the validation conclusion about the validity of OMPS retrievals above **15 km** based on comparison with ozonesonde measurements.

-    This paper should discuss the difference of comparison results between 2016 and 2013. The comparison with MLS provide same results between 2016 and 2013?

**The following is the minor suggestions for technical corrections (I just suggest a few)**

1) Please change "facilitate, overarching, exploit" to more proper words.
2) Many sentence is unnecessarily formatted like "very long subject" + "passive verb".
   e.g) ozone concentrations in the 12-60 km altitude range can be retrieved ➜ ozone concentrations can be retrieved from 12 to 60 km with valid precisions.
   e.g) Observation at altitude where the measurement are contaminated by clouds are rejected by applying a cloud filter ➜ We screen out cloud-contaminated measurements using the color Index ratio of the radiance at 754 and 997 nm.
   e.g) the following molecular specifies with spectral signatures in the selected spectral ranges are considered. ➜ The radiation calculation take account of NO2 and O4 other than ozone.
   e.g) ozonesonde data from WOUDC and SHADOZ archives are used in this analysis ➜ ozonesonde data is collected from WOUDC and SHADOZ archives.
3) Line 3, page 1: SCIAMACHY instrument ➜ SCIAMACHY limb instrument
4) Line 10, page 1: Results for seven months ~~ ➜ OMPS ozone profile retrievals are validated against both satellite-based and balloon-borne measurements for seven month from July 2006 to January 2007.
5) Line 14, page1: those from ozonesondes ➜ ozonesondes or ozonesonde measurements
6) Line 23, page 1 : a stratospheric ozone layer -> the stratospheric ozone layer
7) Line 24, page 2: result in the depletion of stratospheric and mesospheric ozone ➜ lead to the destruction of stratospheric ozone.
8) Line 25, page 2: both from ground-based instrument and satellite observations ➜ from both A and B.
9) Line 34, page 2: the former instruments point downward while the latter look directly into the solar disk : "whereas" is better than "while"
10) Line 35, page 2: The same geometry of observation can also be ➜ has been
11) Line 1, page 3: ~ limb emission measurements. With this latter technique a day and night

coverage of the globe is feasible. ➔ limb emission measurements can be taken during both day and night.

12) Line 5, page 3: launched in March 2002 ➔ launched in March 2002 on board the ESA ENVISAT satellite. Line 7 page 3: In early 2012 ground communication with the ESA ENVISAT satellite, carrying SCIAMACHY among other ozone science relevant instruments, was lost ➔ SCIAMARCHY ended its operation in early 2012 due to the loss of their platform with ground communication.

13) Indents when a paragraph changes. e.g in the lines 3, 22 on page 2,  14line on page 6

14) Edit the usage of reference: e.g line 5 on page 3, (Burrows et al. (1995, Gottwald and Bovensmann (2011))➔ (Burrows et al, 1995;  Gottwald and Bovensmann, 2011). These unformatted types are often found in this article.

15) Lines 11-13, page 3➔  This paper presents ozone profile retrievals from OMPS limb observations. This algorithm was adapted from the SCIAMACHY v3.0 ozone retrieval algorithm (Jia et al., 2015) developed by the University of Bremen.

16) Line 13, page 3: For a description of SCIAMACHY v3.0 ozone retrievals refer to Jia et al. (2015) ➔  readers are referred to Rodgers [2000] for more detailed description of ~.

17) Line 14, page 3: delete "of this paper" after In sect.2

18) Line 16, page 3: The applied cloud filter, the retrieval of aerosol extinction profiles and of the surface albedo ➔  The applied cloud filter and the retrievals of aerosol extinction profiles and surface albedo

19) Line 20, page 3: In the latter section and in the conclusions ➔ in the conclusions

20) Line 21, page 3: OMPS-LP is not mentioned in the introduction before the title name of OMPS-LP instrument.

21) Line 27, page 3: A Nadir Mapper, a Nadir Profiler and a Limb profiler (LP) => the Nadir Mapper, Nadir Profiler, and Limb Profiler.

22) Line 9, page 5: slower that  ➔ slower than

23) Line33, page 7: positive angles are East of the North : change from "are" to "represent"

24) Line 11, page 9: get rid of ➔ remove

25) Cross section of these gases are respectively taken from , respectively.

26) Line 18-19, page 9: delete "used in the radiative transfer mode" and "provided by the NASA team together with OMPS-LP L1 radiances"

27) Line 8, page 14: the geographic distance is required to be whine 1 deg. ➔ limited to be

28) Line 15, page 14: The number is in the order of 5000. ➔ The number is ~ 5000.

29) Line 1, page 15: ➔ the positive difference of larger than 30 % in the tropical lower stratosphere.

30) Line 15, page 15: Looser collocation criteria than for MLS ➔ compared to MLS

31) Line 16, page 15: because of the sparseness of the data set ➔ because of the sparseness of ozonesonde station. / In particular ➔ Therefore

32) Line 18, page 15: remove "generally for each sonde profile ~ found using these loose criteria"

33) Line 4, page 16: with respective standard deviations ➔ with corresponding standard deviations.

34) Line 14, page 16: for tropical and northern mid-latitude bands, around 120 and 160 sonde profiles, respectively are considered. ➔ , which is ~ 120 and 160 for tropical and northern mid-latitude bands, respectively.

35) Line 1, page 17: As can be seen also from Fig.11 ➔ As shown in Fig. 11, the excellent agreement is also found at northern mid-latitudes, with ~~.

---

## Referee Comment (RC2) · Anonymous Referee #2 · 29 Sep 2017

==== General comments

This is a nice paper that does a good job of introducing a new OMPS-LP retrieval approach and describing the dataset resulting from it. I see this paper as ideally suited to the AMT journal and a welcome addition to the body of literature. My comments are all pretty minor and mainly involve requests for further clarification or suggestions of wording changes etc. I'm confident that, once these are addressed, the paper will be ready for publication.

Before I provide some line-by-line comments and suggestions, just a few "global" thoughts. In several places the paper presents comparisons between the IUP-OMPS and another dataset without (that I could readily find) being completely explicit about whether it's  minus <other> dataset that's being presented (as I'm pretty

sure it is) or the sign is reversed. Furthermore, when a percentage or relative difference is shown, you should be clear about what is in the denominator, is it IUP-OMPS, the other dataset or some combination of the two?

The abstract and introduction talk about this paper setting the stage for a potential "combined" dataset linking this new record to the SCIAMACY observations. It would be useful to return to this point in the conclusions section and briefly discuss the consequences of your findings for such an activity. Which of the factors uncovered in this analysis might present challenges to such data fusion?

Finally, I'm aware of at least on other team developing an OMPS-LP data record, that being the OSIRIS team in Saskatoon. Depending on the availability of data from that team, it's worth considering the possibility of expanding section 4.1 (or adding a new section) that at least discusses their approach and its similarities and differences from yours and the NASA one, and perhaps even performs an additional data comparison if appropriate.

==== Specific comments

—- Abstract

It would be good to spell out SCIAMACY and MLS in the abstract (if space permits)

—- Page 1

Line 20: Odd wording of 2nd sentence. How about "... in the atmosphere. It is most abundant in the stratospheric 'ozone layer', which absorbs..."?

—- Page 2

Line 18: For completeness, I suggest you add discussion of the GOZCARDS (doi:10.5194/acp-15-10471-2015) and SWOOSH (doi:10.5194/acp-15-10471-2015) datasets also.

—- Page 3

Line 8: "satellite missions" -> "satellite instruments"

Line 25/26: This needs rewording. First, OMPS is an instrument not a mission (the Suomi NPP missions has "stated aims" that go far beyond ozone). Secondly, while the OMPS-LP and OMPS-NP components are indeed focused on the vertical distribution, you've neglected the OMPS-NM mapping capability which has no vertical resolution and thus a different science focus.

—- Page 6

Lines 1/2: "further prior handling" is odd wording (further and prior sound contradictory), how about "additional screening or processing" or something similar?

—- Page 7

Line 3: Give a citation or more details on the "another scene-based technique".

Lines 7/8: This reasoning doesn't actually quite follow. Photons from any altitude can be scattered within the instrument to any other altitude. It so happens that there are more photons in the lower atmospheric views than the upper atmospheric one. The way it's currently written makes it sound more one way than theoretically can be (though granted, you do start with "For example").

Line 23: "In the preparation time of this paper" -> "At the time of writing this paper"

Line 33: Perhaps delete "the" before "North"?

—- Page 8

Lines 10-15: Please be explicit about whether "approximate spherical" is referring to the assumed shape of the Earth (as I assume it is) or to the shape of scattering particles. How does "approximate spherical" (line 12) relate to "pseudo-spherical" (line 14, page 9 line 1). Also how is all of this related to the oblateness of the Earth, are you assuming a spherical Earth surface but with a radius tuned to give approximately the same shape as the Earth ellipsoid along the line of sight?

—- Page 9

Lines 20-24: Please give more details on what this "shift and squeeze" is correcting (some instrumental anomaly?) and why this correction is necessary (also why it is not needed in the UV range).

— Page 10

Line 12: Typo with Tikhonov

Also line 12: If gamma linearly increases with height then it's a vector rather than a scalar surely (or even possibly a diagonal matrix). Please clarify.

Line 24: Insert "Level 1" after "normalized"?

Line 27-29: I'm not quite sure I understand this. It seems like you're preselecting which wavelength/height subsets of the Level 1 data to use based on the strength of the weighting functions. However, the retrieval factors those strengths in when deciding how much attention to pay to each individual measurement anyway. Why is this additional step, which, in effect, second guesses the retrieval, needed? If including the "weaker" signals has undesirable effects on the result, is it understood why that is? Also, this means that, potentially, each ozone profile was generated by a different "subset" of the instrument, making for a measurement dataset whose properties (precision, resolution etc.) are a moving target, complicating the development of average datasets, long term records, etc. Some discussion of the size of these effects would be good.

—- Page 11

Line 5: suggest "... to reject THs <with radiances> affected by ..."

Line 14: Insert "liquid" before "water"?

—- Page 12

Line 5: Start of line: "Aerosol extinction..." -> "An aerosol extinction..."

Line 6: "... has a coarser spectral resolution <than SCIAMACHY>, ..."

—- Page 13

Line 1: "downwards from" -> "below"

Line 15: "At the moment of submission of the paper" -> "At the time of writing"

Line 17: "... Fig. 8, which shows relative ..."

—- Page 14

Line 2: "AURA" -> "Aura"

Line 4: "satellite suite" -> "MLS instrument"

Line 8: Is there a reference or definition for "modified potential vorticity".

Lines 11-14: Please state what temperature/height information is used to do the density/height to pressure/vmr conversion?

—- Page 15:

Line 9: "... related to impacts of polar mesospheric clouds on the signals that were not successfully screened out of the Level 1 data" or similar wording?

—- Page 17

Line 16: "about" -> "into"

—- Figure 1

Wouldn't hurt to define TH, TP in the caption.

— Figure 2

Again, make figure more "stand alone" by defining "TP"

— Figures 8 and on: Be clear in each what the sign of the differences shown are. (Do it in both the body text and the figure/caption to allow the figures to "stand alone")

---

## Author Response (AR1)

**Author's response to the Referees comments on the manuscript 'Retrieval of ozone profiles from OMPS limb scattering observations' by C. Arosio et al.**

We thank the reviewers for the time they spent carefully reading the manuscript and constructively commenting on the paper. In the text below, referees' comments are shown in italicized font and authors' responses are highlighted in blue.

**1 Anonymous Referee #1**

**General Comment** The overall parts of the paper are written unclearly and illogically. For example, vertical grids where OMPS ozone profiles are retrieved and the unit of ozone should be introduced in the algorithm description section, at once. However, I need to search from. -In line8 on page 9, the vertical range between 12 and 60 km

-In line3 on page 10, unit of ozone: VMR

-In line13 on page 14, authors described "MLS are converted from VMR vs. pressure into number density vs. altitude, interpolated at the regular altitude grid of OMPS", in addition, the regular altitude grid is not mentioned before.

-In line 10 on line 16, 2.6 km corresponding to an average vertical resolution of the retrieval scheme.

We agree with the reviewer's comment, the information has been consolidated and put in the 'Algorithm implementation' section together with the altitude grid information. In SCIATRAN the state vector is used in terms of VMR (because the shape of the VMR profile is more suitable for use with smoothing constraints), whereas the retrieval results are provided in terms of both number density and VMR as a function of altitude. We choose to perform the comparisons with the other data sets in terms of number density, because the uncertainty on the number density profile is smaller (due to a less sensitivity to the temperature profile). In addition, plotting profiles in terms of number density is more interesting for the comparison with ozonesondes. The average resolution of 2.5 km is not related to the retrieval grid but to the resolution of the retrieved profiles as computed using AKs and as shown in Fig. 7 (old Fig. 6).

This article should be checked line-by-line to become more scientific. For example, in abstract, authors mentioned "ozone in the 12-60 km can be retrieved due to using spectral window in the Hartley, Huggins and Chappuis ozone absorption band" In the view of the spectral window, this instrument is optimized to detect ozone over the troposphere including surface rather than the stratosphere. Limb measurements has lack sensitivity to troposphere due to its viewing geometry.

Looking at the weighting functions of ozone at different wavelengths, one sees that the Hartley band is appropriate to retrieve ozone in the upper stratosphere, Huggings (305 and 330 nm) in the middle stratosphere and Chappius band in the lower stratosphere and troposphere (as it is also shown in the NASA's ATDB document pg. 34). In accordance with the reviewer's comment about the lack of sensitivity of limb measurements to troposphere, a statement has been added in the introduction: 'With decreasing altitude the atmosphere becomes more opaque, which results in a decreasing sensitivity of the limb-scatter measurements in the troposphere.'

In addition, authors described the OMPS-LP official algorithm as an inversion scheme with a priori constraints and a Tikhonov regularization, but in OMPS documentation, it is based on an optimal estimation based regulated by a set of a-priori constraints. These two schemes are

not same.

Thanks for the remark, we agree with the reviewer and have changed the manuscript text in accordance.

Please change "Level 1" to "Level 1b" because these two product are not same.

The notation L1G has been introduced, i.e. Level 1b gridded data, in accordance with the NASA's notations.

Insufficient analyses on retrieval/validation were performed, which is commented in the main comment section. I found the text not be precise enough concerning unformatted types, grammatical error, English usage, which is commented in the minor comment section.

Comments in the main section as well minor comments have been addressed as described below. English has been proofread.

**The following is the main suggestions for improvements.**

**0. Abstract**

- Remove "this algorithm was originally developed  $\sim \sim$  to produce a combined data set" in the abstract part and add more about the retrieval related description or results. For example, the vertical resolution of retrievals vary from  $\sim 2.5$  km at lower altitude levels ( $< \sim 30$  km) and  $\sim 1.5$  km to upper altitude levels (from 40 km to just below top levels). The theoretical retrieval precisions are estimated to be 1-5 % above 25 km, but rapidly increase to 15 % at 20 km.

In our opinion this statement provides an important introduction about the motivation of this study. This is why it has been kept. As suggested, additional information was added in the abstract about the retrieval characterization.

- "The optimization of the retrieval algorithm  $\sim \sim . \rightarrow$  This algorithm use altitude-normalized radiances in the UV and VIS wavelength range.

The sentence has been accordingly changed as: 'The retrieval algorithm uses altitude-normalized radiances in the UV and Vis wavelength ranges to obtain ozone concentrations in 12-60 km altitude range.'

- indicating a good agreement  $\rightarrow$  specify the altitude range showing a good agreement e.g.) a demonstrating a good agreement from 15 km to 58 km.

Some details about what 'good agreement' means are already in the following sentences: so 'indicating a good agreement' was deleted.

- did not mention about the comparison with OMPS/NASA product.

Added without details. Now the sentence, considering also the previous comment, is: 'OMPS ozone profiles are retrieved for seven months, from July 2016 to January 2017. Results are compared with NASA ozone profile product and validated against profiles derived from passive satellite observations, or measured by balloon-borne in situ sondes.'

**1. Introduction**

- Authors mentioned that the main objective of the study is to create the long-term dataset using OMPS and SCIAMARCH. To do this, how to overcome the discrepancy of two instrument calibration? It is very difficult because of little overlapping period between OMPS and SCIA-MARCH. Please add shortly how to overcome the discrepancy of two instrument calibration.

The MLS measurements are planned to be used as transfer function to overcome the calibration discrepancy. A corresponding paragraph was added in the conclusion, as also suggested by the other reviewer: 'In light of the results presented here, an additional work for tuning of some retrieval settings is needed before processing the whole data set and attempting the merging with the SCIAMACHY time series. Since the same 1-D retrieval approach has been used for both data sets, we expect this to ease the merging. Unfortunately, only a couple of overlapping months between the two instruments are available, so that a third product must be used for the merging. After the good agreement found in the comparison of our retrievals with MLS, we are considering the use of the latter instrument as a transfer function to handle calibration issues in the merging procedure.'

- Authors too much simplified the summery of the previous studies related to your data product compared to the history and importance of ozone chemistry. It might be better to remove the ozone chemistry-related part (this part is unclearly written) and to focus on 1) history of satellite ozone observation using limb instrument, 2) why we need limb instrument compared to nadir instrument for ozone observation 3) why we need solar scattered limb measurements compared to infrared/microwave emission limb measurements for ozone observation, 4) history of SCIAMARCHY limb ozone profile product; algorithm development/validation, the long-term stability of both instrument and ozone dataset, 5)OMPS LP ozone profile product from OMPS science team at least and others if possible (e.g. Daniel et al. 2017 recently submitted to AMT), 6) the effort of this study to optimize the SCIAMARCHY algorithm for OMPS.

Following the reviewer's suggestions, the chemistry-related part has been reduced and more details added about limb observations and OMPS LP products. In our opinion it is not important to explain the history of SCIAMACHY ozone product in this paper, since it would be off-topic.

- Line 33-35, page 2: the limb combines the advantage of the other two techniques  $\sim \sim$  with relatively high vertical resolution and horizontal coverage; reader who have no idea about satellite instrument could be confused that which instrument has higher vertical (horizontal coverage) resolution compared to Limb.

The sentence has been rewritten to avoid misunderstanding: 'The limb sounding technique, widely used by more recent satellite instruments, combines the advantage of these two: the long path through the atmosphere provides a high sensitivity to trace gases and the variation of the observation angle enables a better vertical resolution with respect to the nadir geometry, featuring a much higher horizontal sampling as compared to the occultation measurements.'

**2. OMPS LP instrument**

**2.1 General features.**

- Line 25 (page 3) "The main objective of the mission is to monitor the ozone vertical distribution within the Earth middle atmosphere at high accuracy level"  $\rightarrow$  it is not true because the mission mentioned belongs just to the SNPP.

Corrected: 'The main objective of OMPS-LP is  $\sim\sim$  '

- Move line 23-27 to introduction and focus just on OMPS LP.

**Done.**

- line 1, page 5: the spectral range between 290 nm and 1000 nm  $\rightarrow$  the spectral range of 290 nm to 1000 nm.

Modified to: 'the spectral range of 280–1000 nm'.

- line 5-8, page 5: The use of such a technology (observation at the same time without vertical scanning and CCD) pose a great challenge as regards the SNR; indeed, scattered solar radiance from the Earth limb decreases by at least five orders of magnitude along the considered vertical range, due to the decrease of atmospheric density.  $\rightarrow$  It is illogically written, about the cause-and-effect.

The sentence has been reworded to be more logical and clear: 'The use of such a technology [CCD] poses a great challenge as regards the dynamic range: indeed, due to the decrease of atmospheric density, scattered solar radiance from the Earth limb decreases by at least five orders of magnitude along the considered vertical range.'

**2.2 Calibration and main issue.** -This part should be simplified or removed and then move some parts in other sections. Example, 1) In algorithm description, we can delivery some calibration issues related to the treatment of this algorithm to overcome these issues 2) In lines 8-9 on page 15, authors mentioned the disagreement between OMPS and MLS can be partly related to pointing issues, due to the solar heating of the instrument at high latitudes or stray light in section 4.2. In this paragraph, this paper can provide more detailed calibration issues related to this discrepancy.

The section was simplified. We currently don't use any other pre-processing steps related to pointing issues in our algorithm and we didn't split this section into the 'algorithm description' and 'MLS comparison' ones to avoid confusion.

-Line2 on page 6: Delete "Level 1B data are provided by NASA team" because the data is publically available.

**Deleted at this point.**

-Line 24 on page 7: delete "In the preparing time of this paper the new data version was not fully released and only seven consecutive months were available." This kind of sentence is not suitable in the scientific article. Move or re-mention "Retrievals were performed using data from the central slit of the instrument only because the lateral slits can still suffer from pointing issues" in the algorithm description or in the beginning of 4. Results.

It was deleted at this point and the sentence has been reformulate at the end of the section, where the data version is introduced, and in the 'Algorithm implementation' one. The expression 'at the time of writing this paper' has been kept since it is related to the chosen period of time and we don't find it inappropriate.

**2.3 OMPS-LP geometry of observations.**

-line 31 on page 7: Azimuth angles could be defined separately as solar azimuth angle and satellite azimuth angle.

For the algorithm only the difference between the two azimuth angles matters. We don't see the need of two separate definitions.

-line 34 on page 7: positive angles are East of the north, so that values are inside the -180 to 180 range  $\rightarrow$  it is hard to understand this sentence.

'so that values are inside the -180 to 180 range' deleted: not necessary detail.

-Why this paper need this section? The information given in this part is never mentioned in other sections.

We kept the section as the figure shows the latitude coverage of the data set in different seasons and it might be useful to characterize the possible influence of the stratospheric aerosol which is strongly related to the scattering angle. A reference to it has been added also in the aerosol section.

**3. Retrieval method**

**3.1 The retrieval algorithm**

-Describe the theoretical inversion scheme first including from line 25 on page 9 to line 18 on page 10, generally and then describe how this algorithm prepare the measurement vector, measurement error vector, forward model vector, and state vector, it might be better to describe them in separated two sections.

The section was re-organized as suggested into two sections: 'Theoretical basis' and 'Algorithm implementation'.

-Move the retrieval characterization and error analysis including Figure 6 in section 4.1 with the changed section title from 4. Satellite data set comparison to 4. Results; 4.1. Retrieval Characterization and Error Analysis 4.2 Comparison with OMPS-LP Ozone Product 4.3 Comparison with MLS 4.4 Comparison with Ozonesonde.

Thanks, this helps the readability. This part was re-organized as suggested.

This study described that "The information content of the measurements as well as the sensitivity of the retrieval can be analyzed using  $\sim \sim$  and the covariance of retrieval noise". It is true for AK, but not true for retrieval error. Sm is generally called "solution error covariance" including random-noise retrieval error covariance and smoothing error covariance. It should be detailed in the paper and an example should be presented in the right panel of Figure 6. It is useful to add the retrieval characterization and error analysis for mid/high latitudes due to the dependence of the sensitivity of solar measurements on solar zenith angles.

In the right panel of Fig. 6 there is already an example of the solution error covariance due to measurements noise, and more examples at different solar zenith angles have been provided. Following (Clarmann, 2014) the smoothing error should not be included in the retrieval error budget.

- The DFS and solution errors of OMPS LP seems to be much better than OMI UV nadir

viewing sensors in the troposphere (Liu et al., 2010). If it is true, we should use OMPS LP measurements for tropospheric ozone retrievals, but it is know that the limb measurements has lack sensitivity to lower troposphere, due to its viewing geometry. I think that the DFS and Retrieval errors are over/under estimated.

We don't retrieve ozone in the lower troposphere, Fig. 6 vertical axis starts indeed at 12 km. Looking at the paper (Liu et al., 2010), the AK peaks in the stratosphere are actually slightly higher in our case but the relative precision is comparable with the OMI one or worse in the lower stratosphere.

-The definition of normalized radiance is unclear  $\rightarrow$  Measurement vector is defined as the logarithm of the altitude-normalized radiances to an upper TH for canceling calibration errors and reducing the effect of surface/cloud reflectance. Table 1 summaries  $\sim \sim$ . In this paragraph, this paper should mention that this algorithm rejects the wavelength between 580 and 670 nm and between 620 and 630.0 to remove the effect of water vapor and O2 absorption when you describe which wavelengths are implemented in this algorithm.

We added equation 4, that explicitly shows the measurement vector.

The cutting of the wavelengths was moved as suggested close to the definition of the chosen spectral ranges.

- Describe that ozone profiles are retrieved at which vertical grids; the number of levels, the vertical intervals, the unit of the grid in the same paragraph.

Done: 'The altitude range over which the retrieval is performed spans between 12 and 60 km above the sea level. The vertical grid is fixed throughout the processing and covers the retrieval range at evenly spaced steps of 1 km.'

-Authors described that ozone retrievals are retrieved from 12 and 60 km in the all sections, but analyzed the retrievals from surface and 60 km.

We never show or discuss results at altitudes below 12 km.

-Line 20-24, page 9: "A shift and squeeze correction is applied in the Chappuis band to the modeded spectrum with respect the measured one: this pre-processing is performed for each observation at each TH independently"  $\rightarrow$  a. describe why the wavelength calibration is implemented just for VIS wavelengths. b. Probably the modeled spectrum is high resolution solar reference data?

a) Sentence added to the paper: 'As the shift and squeeze correction algorithm works with the differential absorption structures, it cannot be applied in the UV range. Furthermore, as the UV retrieval uses either radiances themselves or their slopes, the influence of a possible spectral misalignment is rather small.'

b) No, we mean just the spectrum simulated with the forward model.

- line 23-25, page 10:  $\rightarrow$  surface albedo is simultaneously retrieved with ozone using two spectral fitting windows ( $\sim \sim$ ) where ozone absorption is weak.

We still mention the usage of sun-normalized radiance, since it is a new feature of v2.5. Sentence reformulated: 'Surface albedo is simultaneously retrieved with ozone using the sun-normalized

radiance provided in the L1G data. Two spectral fitting windows at THs around 38 km are employed: 355-365 nm and 455-470 nm, where ozone absorption is weak. '

**4. Satellite data set comparison 4.1 NASA retrieval and comparison**

- Line 15: "At the moment of the submission of the paper, only version 2 of Level 2 (L2) NASA product was available, so a comparison with the most recent retrieval could not be performed" This description is not suitable. This study should use the version 2.5 or should confirm from OMPS science team that there is insignificant difference between v2.0 and v2.5 product. This paper mentioned that OMPS/NASA algorithm is based on an inversion scheme with a prior constraints and a Tikhonov regularization, which should be changed to "an optimal estimation based regulated by a set of a-priori constraints".

Only recently v2.5 L2 daily files have been produced and are now available, covering the period 2014-2017. As a consequence, following the reviewer's comment, Fig.8 has been updated, using v2.5 L2 data of NASA.

As above mentioned the explanation of the algorithm has been changed.

-Based on Figure 8, there are significant differences between OMPS/NASA and OMPS/IUP products, which different implementations between algorithms causes these differences? Based on Figure 9, it seems that MLS shows better agreement with OMPS/IUP in the stratosphere (ozone peak layer) and with OMPS/IUP in troposphere. Both OMPS and MLS has lack sensitivity to lower troposphere so the retrievals determine mostly from a priori information, the similarity between two product might come from the similarity of a priori data between two algorithms.

As the retrieval implementations are different, biases at specific altitudes can not directly be linked to the algorithm differences. It is also impossible to "switch off" the differences step by step as most of the "mixed" algorithm version will not be stable and able to produce any reasonable results. Yes, the comparison between satellite data sets in the upper troposphere is difficult due to the lower sensitivity to ozone, as we also state in the paper.

- OMPS/NASA should be compared with MLS and ozonesonde to see which one provides better retrieval qualities.

We think that joint comparison between NASA-OMPS, IUP-OMPS and MLS/ozonesondes is not the target of this paper: NASA's v2.5 just became partially available and our retrieval is still in progress.

**4.2 MLS comparison**

-change the reference of Waters et al. (2006) to MLS v.2 data quality and description documentation. This doc specifies how to use MLS product as following. This study use this data screening method?

The reference has been added rather than replaced. Yes, the flags reported in this document were used in the comparison between the two data sets: a corresponding sentence has been added.

- In this section, we firstly give a description of the vertical grid and the unit of ozone profile

used in comparison, but this part should be moved before comparison with OMPS/NASA. I think that this paper create one section to describe the comparison methodology.

The vertical grid has been now described in the retrieval section. However, as the comparison methodology slightly differs for different comparisons it was not moved to a dedicated section.

-This paper mentioned "an increase of the smoothing parameter is expected to partially attenuate the latter problem", about the large difference between OMPS and MLS profiles around 50 km. This explanation is so vague. Smoothing parameter indicates smoothing errors?

Smoothing parameter means Tikhonov parameter (changed): we were just addressing the oscillations seen at the top levels (58-60 km), not the one around 50 km.

- Figure 10 could be re-analyzed for several months (July and Dec or summer and winter) due to sufficient collocation.

The original plot has been kept but 2 other plots for summer and winter months were added.

-This paper can mention about the validity of OMPS retrievals above  $\sim 15$  km and below 58 km based on comparison with MLS.

Added at the end of the paragraph: 'To summarize, this comparison shows a general validity of IUP-OMPS retrieval between 18 and 58 km, even if during different season the relative bias with respect to MLS exceeds by 10 % in some limited atmospheric regions.'

-Line 4 page 14: What is the modified potential vorticity?

The adjective 'modified' has been deleted.

-Line 9 page 15: "not screened polar mesospheric clouds"  $\rightarrow$  based on the cases provided in this paper, it is hard to relate the large difference between OMPS and MLS to polar mesospheric clouds (PMC). That is because the presence of PMC is limited to polar summer season, but your analysis is performed for all seasons. This article did not mention that why the presence of PMC is important for OMPS retrievals and why MLS could be not impacted by PMC, maybe need some reference.

Thanks, this plot has been changed after the implementation of a PMC flag, consequently a short paragraph has been added in the Cloud Filter section, addressing the issues and the flagging. The reference to (Bak et. al 2015) paper was added: the authors use MLS as a reference when OMI detects PMCs.

**5. Ozonesonde comparison**

-Convolution process of higher resolution profiles with averaging kernels could be described after equation (4).

We think that moving this to the retrieval section would lead to much more confusion: we don't use smoothing in other parts of the paper.

-This paper mentioned Figure 12 (a) as "averaging kernel smoothing and (b) as "vertical averaging". Please correct this way to "Comparison of OMPS ozone profiles with ozonesonde

smoothed with OMPS averaging kernel and (b) without smoothing, respectively".

Also the Panel (b) shows smoothed profiles but instead of using the AK to smooth the high resolution sonde measurements, a direct vertical averaging over a range of 2.5 km was performed (kind of box-car averaging kernels).

-This paper can add about insignificant impact of the smoothing of ozonesonde profiles to OMPS vertical resolution on the comparison results in the stratosphere due to the comparable vertical resolution of OMPS LP ozone profile retrievals to ozonesonde, compared to the comparison between nadir UV ozone product and ozonesonde. This fact can emphasize the importance of limb instrument on the stratospheric ozone observation.

A sentence related to panel (a) and (b) of Fig. 12 has been added: 'Differences between the two panels of this figure show that the smoothing procedure can be critical in the comparison between 15 and 20 km, where the gradient in the ozone profile is usually strong'. We did not stress here the point related to the better resolution of limb sensors in comparison with nadir ones, because not on-topic. In addition, the difference of resolution between ozonesondes and OMPS-LP is still large: for sondes it's around 10 m, for OMPS in the order of 1 km.

-Should summarize the validation conclusion about the validity of OMPS retrievals above 15 km based on comparison with ozonesonde measurements.

We stressed the point at the end of the paragraph: 'Concluding, we find a general consistency of IUP-OMPS retrieval results with ozonesonde measurements in all considered latitude bands, except for the 12-20 km altitude range in the tropics, where the agreement with SHADOZ ozonesondes is ambiguous.'

-This paper should discuss the difference of comparison results between 2016 and 2013. The comparison with MLS provide same results between 2016 and 2013?

As stated in the paper the two periods were processed using the same settings. Yes, there are no substantial changes in the relative differences IUP-OMPS - MLS between the 2 periods, a sentence was added.

**The following is the minor suggestions for technical corrections (I just suggest a few)**

1)Please change "facilitate, overarching, exploit" to more proper words.

As the reviewer does not explain why (and where) the words are improper and which words he thinks suit better we did our best to go through all occurrences and use other words instead, if appropriate.

2) Many sentence is unnecessarily formatted like "very long subject" + "passive verb".

e.g) ozone concentrations in the 12-60 km altitude range can be retrieved  $\rightarrow$  ozone concentrations can be retrieved from 12 to 60 km with valid precisions.

e.g) Observation at altitude where the measurement are contaminated by clouds are rejected by applying a cloud filter  $\rightarrow$  We screen out cloud-contaminated measurements using the color Index ratio of the radiance at 754 and 997 nm.

e.g) the following molecular specifies with spectral signatures in the selected spectral ranges are

considered.  $\rightarrow$  The radiation calculation take account of NO2 and O4 other than ozone. e.g) ozonesonde data from WOUDC and SHADOZ archives are used in this analysis  $\rightarrow$  ozonesonde data is collected from WOUDC and SHADOZ archives.

The text has been checked and some sentences changed according to the reviewer's suggestion, to avoid recurrent 'very long subject' + 'passive verb' patterns.

3)Line 3, page 1: SCIAMACHY instrument  $\rightarrow$  SCIAMACHY limb instrument

This statement is incorrect as, unlike OMPS, SCIAMACHY is one instrument working either in limb or in nadir observation mode.

4)Line 10, page 1: Results for seven months  $\sim \sim \rightarrow OMPS$  ozone profile retrievals are validated against both satellite-based and balloon-borne measurements for seven month from July 2006 to January 2007.

Changed as 'OMPS ozone profiles are retrieved for seven months, from July 2016 to January 2017. Results are compared with NASA ozone profile product and validated against profiles derived from passive satellite observations, or measured by balloon-borne in situ sondes.'

5)Line 14, page 1: those from ozonesondes  $\rightarrow$  ozonesondes or ozonesonde measurements

Done.

6)Line 23, page 1: a stratospheric ozone layer  $\rightarrow$  the stratospheric ozone layer

**Line deleted**

7)Line 24, page 2: result in the depletion of stratospheric and mesospheric ozone  $\rightarrow$  lead to the destruction of stratospheric ozone.

The statement mentioned by the reviewer is not present in the indicated line/page.

8)Line 25, page 2: both from ground-based instrument and satellite observations  $\rightarrow$  from both A and B.

**Done**

9)Line 34, page 2: the former instruments point downward while the latter look directly into the solar disk : "whereas" is better than "while"

**Done**

10)Line 35, page 2: The same geometry of observation can also be  $\rightarrow$  has been

**Done**

11)Line 1, page 3:  $\sim$  limb emission measurements. With this latter technique a day and night coverage of the globe is feasible.  $\rightarrow$  limb emission measurements can be taken during both day and night.

Sentence reformulated: 'Using the scattered solar light, measurements during daylight only are possible, whereas, using the emission signatures, observations can be performed during both day and night.'

12)Line 5, page 3: launched in March 2002  $\rightarrow$  launched in March 2002 on board the ESA EN-VISAT satellite. Line 7 page 3: In early 2012 ground communication with the ESA ENVISAT satellite, carrying SCIAMACHY among other ozone science relevant instruments, was lost  $\rightarrow$ SCIAMARCHY ended its operation in early 2012 due to the loss of their platform with ground communication.

Sentence slightly modified: 'SCIAMACHY made observations in the UV, Vis, Near InfraRed (NIR) and Short Wave InfraRed (SWIR) spectral ranges till April 2012, when the platform-toground communication was lost.'

13)Indents when a paragraph changes. e.g in the lines 3, 22 on page 2, 14 line on page 6

**Done**

14) Edit the usage of reference: e.g line 5 on page 3, (Burrows et al. (1995, Gottwald and Bovensmann (2011))  $\rightarrow$  (Burrows et al, 1995; Gottwald and Bovensmann, 2011). These unformatted types are often found in this article.

**References were checked.**

15)Lines 11-13, page  $3 \rightarrow$  This paper presents ozone profile retrievals from OMPS limb observations. This algorithm was adapted from the SCIAMACHY v3.0 ozone retrieval algorithm (Jia et al., 2015) developed by the University of Bremen.

Reformulated as: 'This paper presents ozone profile retrievals from OMPS-LP observations performed at the University of Bremen. The algorithm we use was adapted from the SCIA-MACHY v3.0 ozone retrieval (Jia et al., 2015).'

16)Line 13, page 3: For a description of SCIAMACHY v3.0 ozone retrievals refer to Jia et al.  $(2015) \rightarrow$  readers are referred to Rodgers [2000] for more detailed description of  $\sim$ .

The line was changed accordingly with the previous comment but the citation kept: we want to refer to the SCIAMACHY data set not to the retrieval theory in Rodgers.

17)Line 14, page 3: delete "of this paper" after In sect.2

**Done**

18)Line 16, page 3: The applied cloud filter, the retrieval of aerosol extinction profiles and of the surface albedo  $\rightarrow$  The applied cloud filter and the retrievals of aerosol extinction profiles and surface albedo

Changed into: 'A more detailed characterization of the retrieval procedure follows, including the applied cloud filter and the approach to consider aerosol extinction profiles.'

19)Line 20, page 3: In the latter section and in the conclusions  $\rightarrow$  in the conclusions

**Done**

20)Line 21, page 3: OMPS-LP is not mentioned in the introduction before the title name of OMPS-LP instrument.

In the reviewed version it is mentioned in the introduction.

21)Line 27, page 3: A Nadir Mapper, a Nadir Profiler and a Limb profiler  $(LP) \rightarrow$  the Nadir Mapper, Nadir Profiler, and Limb Profiler.

Done, now in the introduction.

22)Line 9, page 5: slower that  $\rightarrow$  slower than

**Done**

23) Line 33, page 7: positive angles are East of the North : change from "are" to "represent"

**Line deleted**

24)Line 11, page 9: get rid of  $\rightarrow$  remove

**Done**

25) Cross section of these gases are respectively taken from  $\sim \sim \rightarrow$  taken from  $\sim \sim$ , respectively.

**Done**

26)Line 18-19, page 9: delete "used in the radiative transfer mode" and "provided by the NASA team together with OMPS-LP L1 radiances"

'Provided by the NASA team' was kept.

27) Line 8, page 14: the geographic distance is required to be whine 1 deg.  $\rightarrow$  limited to be

**Done**

28) Line 15, page 14: The number is in the order of 5000.  $\rightarrow$  The number is ~ 5000.

**Done**

29)Line 1, page 15:  $\rightarrow$  the positive difference of larger than 30 % in the tropical lower stratosphere.

Changed as: 'Starting the discussion form the bottom of the plots, positive differences larger than 30~% are found in the tropical lower stratosphere.'

30)Line 15, page 15: Looser collocation criteria than for  $MLS \rightarrow$  compared to MLS

**Done**

31)Line 16, page 15: because of the sparseness of the data set  $\rightarrow$  because of the sparseness of ozonesonde station./ In particular  $\rightarrow$  Therefore

**Done**

32)Line 18, page 15: remove "generally for each sonde profile  $\sim$  found using these loose criteria"

This part of the sentence was deleted and reformulated as: 'For each sonde profile, all collocated OMPS-LP observations are averaged before the comparison.'

33)Line 4, page 16: with respective standard deviations  $\rightarrow$  with corresponding standard deviations.

**Done**

34)Line 14, page 16: for tropical and northern mid-latitude bands, around 120 and 160 sonde profiles, respectively are considered.  $\rightarrow$ , which is ~ 120 and 160 for tropical and northern mid-latitude bands, respectively.

**Done**

35)Line 1, page 17: As can be seen also from Fig.11  $\rightarrow$  As shown in Fig. 11, the excellent agreement is also found at northern mid-latitudes, with  $\sim \sim$ .

Done: 'As shown in Figs. 11 and 12, an excellent agreement is found at northern mid-latitudes, with relative differences below 5 % between 14 and 30 km.'

**2 Anonymous Referee #2**

**==== General comments**

This is a nice paper that does a good job of introducing a new OMPS-LP retrieval approach and describing the dataset resulting from it. I see this paper as ideally suited to the AMT journal and a welcome addition to the body of literature. My comments are all pretty minor and mainly involve requests for further clarification or suggestions of wording changes etc. I'm confident that, once these are addressed, the paper will be ready for publication.

Before I provide some line-by-line comments and suggestions, just a few "global" thoughts. In several places the paper presents comparisons between the IUP-OMPS and another dataset without (that I could readily find) being completely explicit about whether it's  minus <other> dataset that's being presented (as I'm pretty sure it is) or the sign is reversed. Furthermore, when a percentage or relative difference is shown, you should be clear about what is in the denominator, is it IUP-OMPS, the other dataset or some combination of the two?

We added an explicit equation (Eq. 8) valid for all the relative differences in the paper and corresponding remarks in figure captions, where appropriate.

The abstract and introduction talk about this paper setting the stage for a potential "combined" dataset linking this new record to the SCIAMACY observations. It would be useful to return to this point in the conclusions section and briefly discuss the con-sequences of your findings for such an activity. Which of the factors uncovered in this analysis might present challenges to such data fusion?

That's an interesting point: we added a couple of sentences in the conclusions: 'In light of the results presented here, an additional work for tuning of some retrieval settings is needed before processing the whole data set and attempting the merging with the SCIAMACHY time series. Since the same 1-D retrieval approach has been used for both data sets, we expect this to ease the merging. Unfortunately, only a couple of overlapping months between the two instruments are available, so that a third product must be used for the merging. After the good agreement found in the comparison of our retrievals with MLS, we are considering the use of the latter instrument as a transfer function to handle calibration issues in the merging procedure.'

Finally, I'm aware of at least on other team developing an OMPS-LP data record, that being the OSIRIS team in Saskatoon. Depending on the availability of data from that team, it's worth considering the possibility of expanding section 4.1 (or adding a new section) that at least discusses their approach and its similarities and differences from yours and the NASA one, and perhaps even performs an additional data comparison if appropriate.

That's true, a mention + citation of this other data set was added, however:

1) Also Saskatoon's paper is currently under discussion, even thought the processing has already been extensively performed.

2) An inter-comparison with this data set is for us behind the scope of this paper, as the Saskatoon retrieval is not yet sufficiently validated and differences because of the usage of 2-D approach are expected. Thus, for now it is unclear how the differences in the results, which are expected to be identified, should be attributed properly.

3) A dedicated paper about retrieval errors using different algorithms is foreseen by NASA team.

==== Specific comments

-Abstract

It would be good to spell out SCIAMACHY and MLS in the abstract (if space permits)

**Done**

— Page 1 Line 20: Odd wording of 2nd sentence. How about "... in the atmosphere. It is most abundant in the stratospheric 'ozone layer', which absorbs..."?

Reformulated as: 'It is most abundant in the stratospheric 'ozone layer', which absorbs...'.

- Page 2 Line 18: For completeness, I suggest you add discussion of the GOZCARDS (doi:10.5194/acp-15-10471-2015) and SWOOSH (doi:10.5194/acp-15-10471-2015) datasets also.

**Added**

— Page 3 Line 8: "satellite missions"  $\rightarrow$  "satellite instruments"

**Done**

Line 25/26: This needs rewording. First, OMPS is an instrument not a mission (the Suomi NPP missions has "stated aims" that go far beyond ozone). Secondly, while the OMPS-LP and OMPS-NP components are indeed focused on the vertical distribution, you've neglected the OMPS-NM mapping capability which has no vertical resolution and thus a different science focus.

Yes thanks, we reformulated it, addressing the sentence just to OMPS-LP: 'The main objective of OMPS-LP is to monitor the ozone vertical distribution within the Earth middle atmosphere at high accuracy level.'

- Page 6 Lines 1/2: "further prior handling" is odd wording (further and prior sound contradictory), how about "additional screening or processing" or something similar?

Reformulated: 'In this paper, version 2.5 of OMPS-LP L1G data has been used without any additional pre-processing related to stray light and pointing.'

- Page 7 Line 3: Give a citation or more details on the "another scene-based technique".

We introduced the other acronym as well. The paper Moy et al. has already been introduced.

Lines 7/8: This reasoning doesn't actually quite follow. Photons from any altitude can be scattered within the instrument to any other altitude. It so happens that there are more photons in the lower atmospheric views than the upper atmospheric one. The way it's currently written makes it sound more one way than theoretically can be (though granted, you do start with "For example").

Yeah, we deleted this part and referred the example to the collection of multiple images on 1 single CCD.

Line 23: "In the preparation time of this paper"  $\rightarrow$  "At the time of writing this paper"

**Done, sentence moved at the beginning of the 'Results' section**

Line 33: Perhaps delete "the" before "North"?

**Done**

— Page 8 Lines 10-15: Please be explicit about whether "approximate spherical" is referring to the assumed shape of the Earth (as I assume it is) or to the shape of scattering particles. How does "approximate spherical" (line 12) relate to "pseudo-spherical" (line 14, page 9 line 1). Also how is all of this related to the oblateness of the Earth, are you assuming a spherical Earth surface but with a radius tuned to give approximately the same shape as the Earth ellipsoid along the line of sight?

Clarified in the paper.

- Page 9 Lines 20-24: Please give more details on what this "shift and squeeze" is correcting (some instrumental anomaly?) and why this correction is necessary (also why it is not needed in the UV range).

Sentences added: 'This pre-processing is performed for each observation at each TH independently and is introduced to account for issues related to the spectral calibration and possible thermal expansion of the detector.' and 'As the shift and squeeze correction algorithm works with the differential absorption structures, it cannot be applied in the UV range. Furthermore, as the UV retrieval uses either radiances themselves or their slopes, the influences off a possible spectral misalignment are rather small.'

- Page 10 Line 12: Typo with Tikhonov

Checked but it was correct.

Also line 12: If gamma linearly increases with height then it's a vector rather than a scalar surely (or even possibly a diagonal matrix). Please clarify.

True, we address gamma as a diagonal matrix.

Line 24: Insert "Level 1" after "normalized"?

Reformulated: 'Surface albedo is simultaneously retrieved with ozone using the sun-normalized radiance provided in the L1G data.'

Line 27-29: I'm not quite sure I understand this. It seems like you're preselecting which wavelength/height subsets of the Level 1 data to use based on the strength of the weighting functions. However, the retrieval factors those strengths in when deciding how much attention to pay to each individual measurement anyway. Why is this additional step, which, in effect, second guesses the retrieval, needed? If including the "weaker" signals has undesirable effects on the result, is it understood why that is? Also, this means that, potentially, each ozone profile was generated by a different "subset" of the instrument, making for a measurement dataset whose properties (precision, resolution etc.) are a moving target, complicating the development of average datasets, long term records, etc. Some discussion of the size of these effects would be good. This was just a description about what was done at the beginning to adjust the spectral ranges used into the retrieval. Once chosen, the settings are then kept fixed throughout the processing. After the review, we decided to delete this paragraph because it can lead to misunderstandings.

- Page 11 Line 5: suggest "... to reject THs < with radiances> affected by ..."

Changed to 'A cloud filter is applied during the ozone retrieval to reject THs at which a cloud is present in the field of view of the instrument'.

Line 14: Insert "liquid" before "water"?

Done

- Page 12 Line 5: Start of line: "Aerosol extinction..."  $\rightarrow$  "An aerosol extinction..."

Changed to 'The aerosol extinction...'

Line 6: "... has a coarser spectral resolution <than SCIAMACHY>, ..."

Done

— Page 13 Line 1: "downwards from"  $\rightarrow$  "below"

We kept 'downward from' because we reject also the TH where the cloud has been detected.

Line 15: "At the moment of submission of the paper"  $\rightarrow$  "At the time of writing"

Sentence deleted: the section has been updated using the V2.5 of NASA L2 data.

Line 17: "... Fig. 8, which shows relative ..."

Changed as: '. Fig. 8 shows a comparison between NASA-OMPS retrievals and our results'.

— Page 14 Line 2: "AURA"  $\rightarrow$  "Aura"

**Done**

Line 4: "satellite suite"  $\rightarrow$  "MLS instrument"

**Done**

Line 8: Is there a reference or definition for "modified potential vorticity".

I just meant potential vorticity, the term came from the ERA Interim extractor we use. 'Modified' was deleted.

Lines 11-14: Please state what temperature/height information is used to do the density/height to pressure/vmr conversion?

It is stated in line 13 'using MLS geopotential height': we are aware of possible trend in this variable but we think that it doesn't have impact on our analysis of few months of data. We are surely going to use ECMWF ERA Interim for the analysis of the whole time series and future merging of the data set.

- Page 15: Line 9: "... related to impacts of polar mesospheric clouds on the signals that were not successfully screened out of the Level 1 data" or similar wording?

Paragraph partially modified: 'Looking at panel (b) we notice that the discrepancy increases: these are months when PMCs are expected. This is an indication of a sub-optimal screening of these clouds...'

— Page 17 Line 16: "about"  $\rightarrow$  "into"

Reformulated as: 'The reasons for this behavior are still under investigation.'

- Figure 1 Wouldn't hurt to define TH, TP in the caption.

Both acronyms were introduced in the caption.

- Figure 2 Again, make figure more "stand alone" by defining "TP"

The acronym was defined in the caption.

- Figures 8 and on: Be clear in each what the sign of the differences shown are. (Do it in both the body text and the figure/caption to allow the figures to "stand alone")

References to Eq. 8 have been added.

**Retrieval of ozone profiles from OMPS limb scattering observations**

Carlo Arosio1, Alexei Rozanov1, Elizaveta Malinina1, Kai-Uwe Eichmann1, Thomas von Clarmann2, and John P. Burrows1

1Insitute of Environmental Physics, University of Bremen, Bremen, Germany 2Karlsruhe Institute of Technology, Karlsruhe, Germany

Correspondence to: carloarosio@iup.physik.uni-bremen.de

Abstract. This study describes a retrieval algorithm developed at the University of Bremen to retrieve vertical profiles of ozone from limb observations performed by the Ozone Mapper and Profiler Suite (OMPS). This algorithm was originally developed for use with data from the SCIAMACHY SCanning Imaging Absorption spectroMeter for Atmospheric CHartographY (SCIAMACHY) instrument. As both instruments make limb measurements of the scattered solar radiation in the ultraviolet and

- 5 visible spectral range, an overarching Ultraviolet (UV) and Visible (Vis) spectral ranges, an underlying objective of the study is to facilitate the provision of obtain consolidated and consistent ozone profiles from the two satellites and to produce a combined data set. The optimization of the retrieval algorithm for OMPS takes into account the instrument-specific spectral coverage by exploiting information from spectral windows in the Hartley, Huggins and Chappuis ozone absorption bands. Thereby, ozone concentrations in the 12 60 retrieval algorithm uses altitude-normalized radiances in the UV and Vis wavelength ranges to
- 10 obtain ozone concentrations in 12–60 km altitude rangeean be retrieved. Observations at altitudes where the measurements are . Measurements at altitudes contaminated by clouds are rejected by applying a cloud filterin the instrument field of view are identified and filtered. An independent aerosol retrieval is performed beforehand and its results are used to account for the stratospheric aerosol load in the stratosphere during the ozone retrieval. Results The typical vertical resolution of the retrieved profiles varies from ~ 2.5 km at lower altitudes (< 30 km) to ~ 1.5 km at upper altitudes (from 40 km to just below top levels).
- 15 The retrieval errors resulting from the measurement noise are estimated to be 1–5 % above 25 km, increasing to 10–15 % below 20 km. OMPS ozone profiles are retrieved for seven monthsof data (, from July 2016 January 2017) are compared to January 2017. Results are compared with NASA ozone profile product and validated against independent data sets from both satellite-based and profiles derived from passive satellite observations, or measured by balloon-borne measurements, indicating a good agreement in situ sondes. Between 20 and 50 km, the OMPS ozone profiles typically agree with the MLS data from the
- 20 Microwave Limb Sounder (MLS) v4.2 results within 5–10 within 5–10 %, with the exception of high northern latitudes (> 70° N above 40 km) and the tropical lower stratosphere. The comparison of OMPS profiles with those from ozonesondes shows an agreement ozonesonde measurements shows differences within  $\pm 5$  % between 14 and 30 km at northern mid-latitudes. At southern mid-latitudes, an agreement within 5–105–10 % is achieved, although these results are less reliable because of a limited number of available coincidences. An unexpected bias of approximately 10 % is detected in the tropical region at all
- 25 altitudes. The processing of the 2013 data set using the same retrieval settings and its validation against ozonesondes reveals a much smaller bias; possible reasons for this behavior are under investigation.

**1 Introduction**

Ozone is one of the most important trace gases in the atmosphere. This is due to its stratospheric layerIt is most abundant in the stratospheric 'ozone layer', which absorbs strong ultraviolet (UV) radiation, heating this atmospheric region and thereby acting as a protective layer against biologically harmful radiation. It is relevant to climate because of its plays a crucial role in the

- 5 radiative budget of the stratosphere. The presence of a stratospheric ozone layer was first discussed by Hartley (1880) but came to prominence in the late 1960s and early 1970s when it was first recognized that anthropogenic activities could result in the depletion of stratospheric and mesospheric ozone (Molina and Rowland, 1974). Ozone is also of importance in the tropospheric ehemistry and it is a greenhouse gas, thus its amount and spatial distribution play a role in the global warming and climate ehange processes (IPCC 5th report, Pachauri and Meyer (2014))., determines the tropopause height and thus also impacts on
- 10 climate. After the recognition that man made man-made release of chlorofluorocarbon compounds depletes the stratospheric ozone layer (Molina and Rowland, 1974) and the discovery of the springtime ozone hole in Antarctica (Farman et al., 1985), the field of stratospheric chemistry and physics researches expanded. This resulted in the mechanism of the production and loss of stratospheric ozone being much better explained. research grew in this field because of its relevance to both science and society. Although nowadays the stratospheric ozone chemistry is generally well understood, there are still several issues
- 15 to be clarified. These are related to the expected ozone recovery after the Montreal protocol adoption, long term ozone trends and adoption of the Montreal protocol, stratospheric responses to changes in tropospheric radiative fluxes and temperatures as well as long term ozone trends. For example, Solomon et al. (2016) focused the attention on the Antarctic region, investigating possible signatures of an ozone healing. Analyzing observations collected each September since 2000, the authors suggested that the fingerprints of an ozone recovery can be identified in both the increase of its column amount and in the decrease of the
- 20 areal extent of the ozone hole.

The issues related to changes in the Brewer Dobson Circulation (BDC), possibly linked to climate changes, are have been investigated by several studies, that which consider the ozone concentration in the lower stratosphere a good proxy to track changes in the stratospheric circulation. Among them, Aschmann et al. (2014) used combined  $O_3$  time series from satellite instruments and ozonesondes to investigate changes in the BDC after the beginning of the century and identified an asymmetry

in the BDC northern and southern branches. Stiller et al. (2017) suggested a shift of the subtropical mixing barriers as an explanation of for this asymmetry.

For all these kinds of studies, reliable long-term data sets are needed from both ground-based and satellite instruments. Recent attempts to consistently merge a large number of different data sets to study trends into long-term time series are reported by Froidevaux et al. (2015) and Davis et al. (2016) both including also other species than ozone. Steinbrecht et al. (2017) and

30 Sofieva et al. (2017) focused on ozone trends, revealing a global statistically significant increase in the ozone its amount after 2000 above 35 km. Other authors, as Kyrölä et al. (2013), Eckert et al. (2014), Gebhardt et al. (2014) and Nedoluha et al. (2015) pointed out an unexpected decadal negative trend in the ozone abundance in the upper tropical stratosphere. The occurrence of strong ozone loss events over the Aretic during spring after particularly cold stratospheric conditions, as occurred in 2011 and 2016, drew the attention of scientists and public concern to the possible consequences for human health (Manney et al., 2011) - Current predictions of a long term impact of global warming coupled with the removal of ozone depleting species indicate a colder stratosphere and an increase in stratospheric ozone, a so-called super recovery of ozone (WMO (2014)). For all these kinds of studies, reliable long-term data sets are needed both from ground-based instruments and satellite observations: for example, recent discussions have shown the importance of multi-decadal time series in order to detect trends in the ozone

5 concentration and the possible recovery of the ozone layer in the Antarctic region (Stolarski and Frith, 2006)

During the last few decades, several remote sensing observation techniques have been used to derive ozone concentrations from the troposphere up to the mesosphere (Hassler et al., 2014). Following the birth of the space age, instrumentation of different kinds began to be developed. Space-borne remote sensing measurements in the Ultraviolet-Visible (UV-VISUV-Vis) spectral range have traditionally been of two types: nadir viewing and solar occultation spectrometers; the former instruments

- 10 point downward while and are characterized by a good horizontal coverage whereas the latter look directly into the solar disk-A more recent technique, the limb scatter of sunlight, combines the advantages of the other twotechniques and provides vertical profiles of ozone density with relatively high vertical resolution and horizontal coverage. The same geometry of observation ean also be exploited to collect measurements, featuring a good vertical resolution and a strong signal. The limb sounding technique, widely used by more recent satellite instruments, combines the advantage of these two: the long path through
- 15 the atmosphere provides a high sensitivity to trace gases and the variation of the observation angle enables a better vertical resolution with respect to the nadir geometry, featuring a much higher horizontal sampling as compared to the occultation measurements. Limb observation geometry has also been used to measure scattered solar radiance and/or atmospheric emission in the InfraRed (IR) or and microwave spectral regions, the so called limb emission measurements. With this latter technique a. Using the scattered solar light, measurements during daylight only are possible, whereas, using the emission signatures,

20 observations can be performed during both day and nighteoverage of the globe is feasible. One of the instruments capable of performing. With decreasing altitude the atmosphere becomes more opaque, which results in a decreasing sensitivity of the limb-scatter observations in the UV, VIS, Near InfraRed (NIR) and Short Wave InfraRed (SWIR) spectral ranges was measurements in the troposphere.

The limb scatter technique was for the first time successfully exploited by the LORE/SOLSE (Limb Ozone Retrieval
 Experiment/Shuttle Ozone Limb Sounding Experiment) instrument launched in 1997 by NASA. Two instruments followed this mission: the Optical Spectrograph and Infrared Imager System (OSIRIS) launched in February 2001 (Llewellyn et al., 1997) and the SCanning Imaging Absorption spectroMeter for Atmospheric CHartographY (SCIAMACHY), launched in March 2002 (Burrows et al. (1995), Gottwald and Bovensmann (2011)). In early (Burrows et al., 1995; Gottwald and Bovensmann, 2011). SCIAMACHY made observations in the UV, Vis, Near InfraRed (NIR) and Short Wave InfraRed (SWIR) spectral ranges

- 30 till April 2012ground communication with the European Space Agency ENVISAT satellite, carrying SCIAMACHY among other ozone science relevant instruments, when the platform-to-ground communication was lost. A few aging satellite missionsinstruments, such as the Optical Spectrograph and InfraRed Imager System (OSIRIS ) OSIRIS and the Microwave Limb Sounder (MLS), are still operating, contributing to the task of continuous monitoring the stratospheric ozone. At the end of 2011, just a few months before the end of ENVISAT lifetime, the Ozone Mapping and Profiler Suite (OMPS) in-
- 35 strument was launched on board the Suomi-National Polar-Orbiting Operational Environmental Satellite System Preparatory

Project (SNPP) platform and it is still operational (Flynn et al., 2014). The spacecraft has a nominal 13:30 local time ascending sun-synchronous orbit and flies at a mean altitude of 833 km. Scientific data collection started at the beginning of 2012. OMPS comprises three instruments: the Nadir Mapper, Nadir Profiler and Limb Profiler (LP). Only the latter is of interest for our study (see Flynn et al., 2014, for a review of the full suite).

5 In this paper the retrieval algorithm developed After the launch of the satellite, the NASA team developed a retrieval chain to derive ozone profiles and many by-products from OMPS limb observations, which are publicly available. Besides, at the University of Bremen to retrieve vertical ozone profiles from OMPS limb observations is discussed. An overarching objective for Saskatchewan a 2-D geometry retrieval has been applied to OMPS-LP measurements (Zawada et al., 2017).

This paper presents ozone profile retrievals from OMPS-LP observations performed at the University of Bremen. The algorithm we use was adapted from the SCIAMACHY v3.0 ozone retrieval (Jia et al., 2015). The underlying objective of

- 10 algorithm we use was adapted from the SCIAMACHY v3.0 ozone retrieval (Jia et al., 2015). The underlying objective of the study is the creation of a consolidated data set and the merging of the OMPS and the SCIAMACHY time series, in order to obtain a long-term continuous data set. For a description of SCIAMACHY v3.0 ozone retrievals refer to Jia et al. (2015) -In Sect. 2of this paper, the OMPS instrument is introduced: its geometry of observation, relevant characteristics and issues related to the retrieval of ozone are briefly discussed. The third section is focused on the retrieval methodology, starting with
- 15 a general description of the inversion algorithm used in this work. A more detailed characterization of the retrieval procedure follows, including the applied cloud filter <del>, the retrieval of and the approach to consider</del> aerosol extinction profiles<del>and</del> of the surface albedo. In . Sect. 4 <del>, presents at</del> first a comparison with NASA ozone profile retrieval algorithm<del>is shown</del>; then MLS and ozonesonde data sets are used for a first validation of our results. Main results, remaining issues and possible future improvements are addressed in the latter section and in the conclusions.

**20 2 OMPS-LP Instrument**

**2.1 General features**

The Suomi-National Polar-Orbiting Operational Environmental Satellite System Preparatory Project (SNPP) platform carrying the OMPS instrument was launched in October 2011 and OMPS data collection started in January 2012. The spacecraft has a nominal 13:30 local time ascending sun-synchronous orbit and flies at a mean altitude of 833. The main objective of the

25 mission OMPS-LP is to monitor the ozone vertical distribution within the Earth middle atmosphere at high accuracy level. OMPS comprises three instruments: a Nadir Mapper, a Nadir Profiler and a Limb Profiler (LP). Only the latter is of interest for our study (see Flynn et al. (2014) for a review of the full suite). OMPS-LP It images the Earth atmosphere by viewing its edge (limb) from space. The closest approach of the sensor line of sight to the Earth surface is referred to as the Tangent Point (TP) and the altitude of this point above the Earth geoid is called Tangent Height (TH); the limb geometry is schematically drawn 30 in Fig. 1.

The OMPS-LP sensor views at the Earth limb backwards with respect to the flying direction, through three vertical slits: the central one is aligned along the nadir track, while whereas the other two are cross-track, separated horizontally by 4.25°, which corresponds to 250 km distance between the TPs. Each slit covers a vertical range of 112, imaging the atmosphere

Figure 1. Schematic diagram of the viewing geometry of a satellite limb observation, showing the so called Tangent Point (TP) and its height above the Earth, or Tangent Height (TH).

without scanning. OMPS-LP was designed to measure ozone vertical distributions in the upper troposphere and stratosphere with an instantaneous field of view of about 1.5 and a sampling of 1 at TP (Jaross et al., 2014). The orbit is inclined, so that the TP is With respect to the satellite, the TPs are located on the Eastwith respect to the satellite, around 25° latitude South of the sub-satellite point. The geometry is drawn in Fig. 2. OMPS-LP The spacecraft completes 14–15 orbits per day and the instrument performs normally 180 limb observations (referred to as states) per orbit, around 160 of which with solar zenith

angle less then 80°<del>, and completes 14-15 orbits per day</del>.

5

---

## Referee Report (RR1)

**General Comments:**

This paper introduces ozone profile retrievals from scattered radiance spectra in the ultraviolet and visible measured by OMPS Limb instrument using a regularized inversion technique. A three kinds of reference data sets (MLS, NASA OMPS-LP O3P, Ozonesonde) are used to assess their retrieval product. The verification results of this product are so interesting and important because this product will be merged with the SCIAMACHY ozone profiles, based on the same algorithm, to create a long-term data set. However, this reviewer would like to comment that the authors should consider deepening strongly the discussion about the OMPS limb Ozone Profile retrievals to convince potential data users of the data quality. Especially, the applied implementations in the retrieval process are mostly adopted from the SCIAMARCHY v3.0 ozone retrieval with small modification, this paper should provide reliable results for the verification of the data product to be published. If not, they well show what big efforts they made to optimize/improve the OMPS limb ozone profile retrievals, different from the original algorithm.

**Detailed Comments:**

1. Page 3, 21-23: Limb observation cannot see below lower stratosphere due to limited field of view and much strong interference with clouds.

2. Figure 7 (right panel): Why the retrieval errors are maximum at minimum solar zenith angles below 25 km and above 45 km in the retrieved altitude range? This retrieval characterization could be related to the maximum errors in lower stratosphere and upper atmosphere over the tropics compared to middle latitudes, shown in all comparison results.

3. This author should demonstrate or intensively discuss that this product have the accuracy/precision at least comparable to NASA OMPS-LP O3P product. This point is most interesting part for data users to determine which dataset they should use. Especially, the comparisons between IUP-OMPS and NASA-OMPS shows a significant bias of 10 % for most altitude, up to 20 % at the bottom level. I think that this difference is very huge considering the products derived from same satellite measurements and a very good vertical resolution of this instrument. This authors provide a detailed description about NASA-OMPS product, but did not discuss why two OMPS limb products have a big difference, especially in the lower stratospheric region over the tropics. So this paper should apply the comparisons with reference dataset to both IUP and NASA OMPS products under the exactly same condition. The NASA v2.5 limb data product is available for the whole period. This comparison could give an insight into the strength/weakness of the retrieval algorithm for a better understanding on the retrievals.

4. Fig 9: The author just introduce the Fig 9, as following, " Fig.9 shows the averaged profiles for the tropics and relative differences in the three latitude bands ", but there is nothing related discussion. Please deepen the discussion about the presented figure, which is corresponding to most figures. Generally, this paper tends to provide a huge description about dada and methodology used in the comparison and very simple/light discussion about the comparison results.

5. Fig 10: The author described that the positive errors above 35 km in NH high latitude during the northern polar summer season are caused due to the presence of the PMC and its sub optimal screening process. If so, why this PMC-induced positive errors are not shown in the SH high latitude during the SH polar summer season (December and January). Based on Bak et al. (2016), OMI UV ozone profiles show systematic PMC-induced errors during both polar summer season and the PMC detection flags systematically works for both Polar areas even though a relatively weak sensitivity of OMI nadir UV measurements compared to limb UV measurements. This IUP OMPS algorithm should be improved in screening the PMC affected pixels because this PMC-induced biases could impact on the long-term data analysis.

6. Comparison with ozonesonde: this paper insist that "the lack of stations presents a meaningful comparison over this short time span or validation is less significant because only two ozonesonde stations are available within the considered time span". If so, this paper should not use the ozonesonde dataset for validating the OMPS dataset or increase the validation period because the OMPS radiance dataset are available for the whole period.

7. Please simply the section 2, more maybe within 1 page, focusing on parts required to introduce this algorithm and to discuss the retrieval results. This part contain 5-6 pages among 23 pages. But, this part is rarely referred in other sections.

8. 12 page: "As the shift and squeeze correction algorithm works with the differential absorption structures, it cannot be applied in the UV range". It is hard to understand because the Huggins ozone absorption bands have notable differential absorption structure.

9. 15p page, 24 line: "The aerosol retrieval is particularly important at latitudes where the scattering angle is high", why ? Please more description using the presented figure 5.

10. In retrieval characterization, this paper just deals with the retrieval errors related to measurement random-noise errors, but discuss the effect of smoothing errors on the comparisons with high-resolution reference dataset. It is not consistent, so it is good to include the retrieval errors related to smoothing errors in section 4.1

**Minor comments**

A few editing correction is suggested and this paper should be more carefully edited.

P1 14L: below top levels : levels => level

P2, 9L : it determines the tropopause height : it is partly true, but the contribution of ozone on tropopause determination is not major.

P2, 14: the discovery of the springtime ozone hole in Antarctica research grew in this field: this sentence is not clearly written.

P13, 13 : each iteration => $i^{th}$ iteration

P14, 6: The CI is defined as the ratio of

P 14, 8: delete "an altitude dependent quantity and "

P15, 3-8: revise this paragraph using "The presence of PMCs can affect limb radiance down to 40 km, causing an interference with ozone retrievals. Therefore, we screen out the PMC contaminated pixels in this study using the PMC detection flag in high latitudes below 50 N and below 50 S where the PMC occurrence is most frequent. PMCS are detected using the radiance profile around 353 nm if the radiance between 40 km and 80 km increases in two consecutive layers at least because radiances decrease monotonically with height in this altitude range under clear sky condition."

P15: 11-13: revise this sentence using "To be optimized for OMPS aerosol retrieval, the wavelength is changed from 750 nm for SCIAMACHY to 868.8 nm for OMPS because the influence of the $O_2$ absorption at 750 nm becomes significant due to the OMPS's coarser spectral resolution.

P16, 8 : with a peak around 35 km ➔ with a worst resolution around 25 km.

Figure: the bottom level of y-axis is marked in all figures.

---

## Referee Report (RR2)

**General Comments:**

This paper has been hugely improved through this revising processes. Now, I found this paper to be well written an organized, and the scientific relevance clearly indicated so I would like to give a minor comment prior to publication.

**Minor comments**

➔ In Figure 10, why the comparison altitude ranges between (a) and (b) is different?

Page 9 Line 6 : ", that" ➔ eliminating "," or changing to ", which" might be better

Page 12 line 8: a correct description of the aerosol scattering is particularly important.

➔ Please better edit.

Page 13, line 12 Please better revise based on has been used, which was improved in aspect to stray light treatment and pointing corrections in comparison to the previous version.

Page 14, line 8: ➔ doublet and triplet methods, respectively

Page 14, line 14➔ Please better revise, it is hard to understand for "With values that follow an approximate linear decrease along the orbit"

Page 15, line 2: at these altitudes the ozone concentration and, thus ➔ the ozone concentration gets very low??

Pape 22, line 19: Please edit " was detected, unexpected"

Page 22: why do you note such like "sonde dataset **were retrieved on** YYMMDD" ?

---

## Author Response (AR2)

**Replies to Referee #1 on the manuscript 'Retrieval of ozone profiles from OMPS limb scattering observations' by C. Arosio et al.**

We thank the reviewer for the time he spent commenting on the paper. In the text below, we address the comments from the referee #1. Referee's comments are shown in italicized font and authors' responses are highlighted in blue.

**==== General comments**

*This paper introduces ozone profile retrievals from scattered radiance spectra in the ultraviolet and visible measured by OMPS Limb instrument using a regularized inversion technique. A three kinds of reference data sets (MLS, NASA OMPS-LP O3P, Ozonesonde) are used to assess their retrieval product. The verification results of this product are so interesting and important because this product will be merged with the SCIAMACHY ozone profiles, based on the same algorithm, to create a long-term data set. However, this reviewer would like to comment that the authors should consider deepening strongly the discussion about the OMPS limb Ozone Profile retrievals to convince potential data users of the data quality. Especially, the applied implementations in the retrieval process are mostly adopted from the SCIAMARCHY v3.0 ozone retrieval with small modification, this paper should provide reliable results for the verification of the data product to be published. If not, they well show what big efforts they made to optimize/improve the OMPS limb ozone profile retrievals, different from the original algorithm.*

Considering a very different spectral resolution of the two instruments and technical differences in the measurement procedure, e.g. in terms of spectral channels, wavelength ranges, atmospheric/scene sampling and radiance collection, it was not possible to apply the SCIAMACHY retrieval scheme to OMPS-LP measurements without significant changes. As a consequence, the algorithm presented in this manuscript has been newly developed starting from the one used for SCIAMACHY. However, the same radiative transfer model (SCIATRAN), a similar retrieval approach and the same spectroscopic and atmospheric parameters databases were used to minimize the systematic errors between the data sets and thus facilitate their merging. We have added a paragraph in the introduction explaining the adjustments of the algorithm in more detail.

To provide a more reliable validation of our product without completely changing the paper structure, we considered the whole 2016 data set, increased the number of latitude bands and updated all the figures in section 4.

Furthermore, we extended the description of the retrieval characterization and of the validation results.

*1. Page 3, 21-23: Limb observation cannot see below lower stratosphere due to limited field of view and much strong interference with clouds.*

The sentence has been reformulated as: 'The accuracy/sensitivity of limb measurements decreases with the altitude in the lower stratosphere and troposphere, as the increasing optical thickness along the line of sight leads to a saturation of the measured signal. The presence of clouds in the field of view acts as an additional limitation.'

*2.Figure 7 (right panel): Why the retrieval errors are maximum at minimum solar zenith angles below 25 km and above 45 km in the retrieved altitude range? This retrieval characterization could be related to the maximum errors in lower stratosphere and upper atmosphere over the tropics compared to middle latitudes, shown in all comparison results.*

Fig. 5 (former Fig. 7) has been updated to show in more detail the latitude (i.e. solar zenith angle) dependence of the relative precision and vertical resolution of the retrieval scheme. We notice the degrading precision in the UTLS region, particularly in the tropics. This is related to the low ozone concentration at these altitudes and latitudes, leading to relative errors up to 10-30%. This purely random uncertainty cannot be directly related to the systematic bias we found below 20 km in the comparison with the other data sets, as the random error is expected to be significantly reduced when averaging the profiles. For example, considering 10000 profiles over the validation period in the tropics and a relative precision of 30 % for each single profile, the random uncertainty on the averaged profile is equal to 0.3 %. However, the large standard deviation in the UTLS region shown in the validation section plots reflects a large variability of the ozone profiles and a lower retrieval sensitivity at these altitudes. This explanation has been added to the retrieval characterization section of the manuscript.

*3. This author should demonstrate or intensively discuss that this product have the accuracy/precision at least comparable to NASA OMPS-LP O3P product. This point is most interesting part for data users to determine which data set they should use. Especially, the comparisons between IUP-OMPS and NASA-OMPS shows a significant bias of 10 % for most altitude, up to 20 % at the bottom level. I think that this difference is very huge considering the products derived from same satellite measurements and a very good vertical resolution of this instrument. This authors provide a detailed description about NASA-OMPS product, but did not discuss why two OMPS limb products have a big difference, especially in the lower stratospheric region over the tropics. So this paper should apply the comparisons with reference data set to both IUP and NASA OMPS products under the exactly same condition. The NASA v2.5 limb data product is available for the whole period. This comparison could give an insight into the strength/weakness of the retrieval algorithm for a better understanding on the retrievals.*

We agree with the reviewer that a thorough comparison with the NASA product would be interesting also for data users. However, the introduction of a triple comparison of IUP and NASA OMPS results with those from MLS, instead of two separate comparisons, would require a complete restructuring of the paper, without contributing significantly, in our opinion, to the scientific value of the manuscript. A dedicated paper devoted to the validation of NASA results and comparison to other OMPS-LP retrievals is currently in preparation by the NASA team and we are also contributing to this work. Including the same results in our paper will cause a double work load without any additional outcome. Since the data quality of NASA product has not yet been assessed in a peer reviewed paper, the reviewer's requirement for our product to 'have the accuracy/precision at least comparable to NASA OMPS-LP O3P product' cannot be directly met at this stage, without performing a full validation of the NASA data set from our side. Furthermore, the main topic of the paper is to present an ozone retrieval algorithm which is optimized for merging with the SCIAMACHY data set. In comparison to the NASA retrieval, this task is achieved to a large extend by providing continuous profiles for the whole altitude range and minimizing systematic errors by using a similar retrieval approach and same data bases for atmospheric and spectroscopic parameters. This is now also clearly stated in the paper. In this respect we do not see us in competition with NASA. This is why we are mainly interested in the absolute accuracy of our retrieval and consider the validation of NASA profiles outside our scientific focus. To our opinion the absolute accuracy of our retrieval is well characterized by comparisons with MLS and sondes: the found agreement within 5-20 %, depending on altitude and latitude, is in a common range achieved in other studies (Mieruch et al., 2012; Tegtmeier et al., 2013; Zawada et al., 2017).

The discrepancies between IUP and NASA profiles, shown in Fig. 6 (former Fig. 8), of about 2-8% above 20 km and up to 20% below 20 km are mainly caused by differences in the used spectral

ranges and in the retrieval approach: NASA algorithm considers spectral points following the triplet method whereas we fit entire spectral windows. In addition, in the NASA product UV and VIS retrievals are kept separated while we try to merge the two spectral ranges around 30-35 km. These facts contribute to a different sensitivity of the retrieval to possible random and systematic errors. Unfortunately, it is not possible to identify and relate each discrepancy at different altitudes to specific settings of the 2 algorithms: it is not feasible to adjust the settings step by step, because the intermediate retrieval versions would result in oscillating or non-converging solutions. From the theoretical point of view, both methods are equally justified and there are no reasons to prefer one to the other, with the exception of the continuity of the IUP profiles.

To meet the reviewer's point about 'comparisons with a reference data set of both IUP and NASA products under the exactly same condition', we considered the same sub-sample of data used for the validation against MLS to produce a plot similar to Fig. 6 (former Fig. 8). That is, only OMPS states collocated with MLS observations were taken into consideration. The reviewer can find the result in Fig. 1. Comparing these two panels with the plots in the paper, no significant differences are found.

[Figure]

Figure 1: Relative differences between IUP and NASA profiles for the Vis (b) and UV (c) retrievals in five latitudinal bands (60° N–90° N, 40° N–60° N, 20° S–20° N, 60° S–40° S and 90° S–60° S). Only the measurements collocated with MLS are considered.

*4.Fig 9: The author just introduce the Fig 9, as following, " Fig.9 shows the averaged profiles for the tropics and relative differences in the three latitude bands ", but there is nothing related discussion. Please deepen the discussion about the presented figure, which is corresponding to most figures. Generally, this paper tends to provide a huge description about dada and methodology used in the comparison and very simple/light discussion about the comparison results.*

A discussion of this figure and a comparison with Fig.7 (former Fig.9) have been added. We did our best to extend and improve the descriptions of the other figures as well.

*5.Fig 10: The author described that the positive errors above 35 km in NH high latitude during the northern polar summer season are caused due to the presence of the PMC and its sub optimal screening process. If so, why this PMC-induced positive errors are not shown in the SH high latitude during the SH polar summer season (December and January). Based on Bak et al. (2016), OMI UV ozone profiles show systematic PMC-induced errors during both polar summer season and the PMC detection flags systematically works for both Polar areas even though a*

*relatively weak sensitivity of OMI nadir UV measurements compared to limb UV measurements. This IUP OMPS algorithm should be improved in screening the PMC affected pixels because this PMC-induced biases could impact on the long-term data analysis.*

We agree with the referee. The PMC detection flag has been updated to optimize the results in the northern hemisphere. We updated Fig. 8 (former Fig. 10) in the manuscript and its description accordingly.

*6.Comparison with ozonesonde: this paper insist that "the lack of stations presents a meaningful comparison over this short time span or validation is less significant because only two ozonesonde stations are available within the considered time span". If so, this paper should not use the ozonesonde dataset for validating the OMPS dataset or increase the validation period because the OMPS radiance dataset are available for the whole period.*

We believe that ozonesondes are a very important validation tool: the high accuracy of these measurements in the troposphere and lower stratosphere is particularly valuable for the comparison of our results over the UTLS region. The sentence quoted by the reviewer referred only to high polar latitudes and mid-latitudes in the southern hemisphere. Considering the extended validation period, the number of available ozonesondes increased significantly and a validation also at northern and southern high-latitudes become possible. We updated the figures and the descriptions accordingly.

*7.Please simply the section 2, more maybe within 1 page, focusing on parts required to introduce this algorithm and to discuss the retrieval results. This part contain 5-6 pages among 23 pages. But, this part is rarely referred in other sections.*

We see the point of the reviewer and already simplified this section during the first revision of the manuscript. We did our best to get it even shorter, now down to 3 pages (including pictures): we think that an introduction of the instrument, its geometry and the issues related to pointing and stray light is important to understand the retrieval implementation and results.

*8.12 page: "As the shift and squeeze correction algorithm works with the differential absorption structures, it cannot be applied in the UV range". It is hard to understand because the Huggins ozone absorption bands have notable differential absorption structure. The corresponding explanation has been added in the manuscript.*

Due to a relatively low spectral resolution of the OMPS Limb Profiler, the differential absorption structure in the Huggins band is largely smoothed out and our algorithm works in the UV range with radiance slope or radiance itself. This explanation has been added to the text.

*9.15p page, 24 line: "The aerosol retrieval is particularly important at latitudes where the scattering angle is high", why ? Please more description using the presented figure 5.*

The aerosol scattering phase function has a strong forward peak, so that a correct description of the aerosol scattering is particularly important at high northern latitudes where the scattering angle is small. The sentence has been modified and corrected in the paper as 'Because of the strong forward peak of the aerosol scattering phase function, a correct description of the aerosol scattering is particularly important at high northern latitudes where the scattering angle is small'.

*10. In retrieval characterization, this paper just deals with the retrieval errors related to measurement random-noise errors, but discuss the effect of smoothing errors on the comparisons with high-resolution reference dataset. It is not consistent, so it is good to include the retrieval errors related to smoothing errors in section 4.1*

In section 4.1 we are not dealing with smoothing errors related to the retrieval algorithm and the Tikhonov regularization constraint. The 'smoothing' procedure is applied to the ozonesondes profiles in order to match the vertical resolution of OMPS profiles. In order to avoid confusion, instead of 'AK smoothing' we now call it 'AK convolution': this is what we indeed perform, a convolution of the high-resolution ozonesonde with the AK of the retrieval scheme. Applying this procedure, the error related to the different vertical resolutions of the compared profiles is accounted for.

**Minor comments** *A few editing correction is suggested and this paper should be more carefully edited.*
The paper has been carefully checked.

*P1 14L: below top levels : levels → level*
The sentence has been partially changed: 'The typical vertical resolution of the retrieved profiles varies from $\sim$ 2.5 km at lower altitudes ($<$ 30 km) to $\sim$ 1.5 km about 45 km and becomes coarser at upper altitudes.'

*P2, 9L : it determines the tropopause height : it is partly true, but the contribution of ozone on tropopause determination is not major.*
The sentence has been changed to: 'It plays a crucial role in the radiative budget of the stratosphere, determines the stratospheric temperature profile and impacts on atmospheric circulation and climate.'

*P2, 14: the discovery of the springtime ozone hole in Antarctica research grew in this field: this sentence is not clearly written.*
The sentence has been has been changed to: 'Because of its relevance to both science and society, ozone-related researches expanded after the discovery of the springtime ozone hole in Antarctica and the subsequent recognition that man-made release of chlorofluorocarbon compounds depletes the stratospheric ozone layer.'

*P13, 13 : each iteration → i th iteration*
Change to 'each i-th iteration'.

*P14, 6: The CI is defined as the ratio of*
Done.

*P 14, 8: delete "an altitude dependent quantity and "*
Done.

*P15, 3-8: revise this paragraph using "The presence of PMCs can affect limb radiance down to 40 km, causing an interference with ozone retrievals. Therefore, we screen out the PMC contaminated pixels in this study using the PMC detection flag in high latitudes below 50 N and below 50 S where the PMC occurrence is most frequent. PMCS are detected using the radiance profile around 353 nm if the radiance between 40 km and 80 km increases in two consecutive layers at least because radiances decrease monotonically with height in this altitude range under*

*clear sky condition."*
The paragraph has been accordingly revised and partially updated.

   *P15: 11-13: revise this sentence using "To be optimized for OMPS aerosol retrieval, the wavelength is changed from 750 nm for SCIAMACHY to 868.8 nm for OMPS because the influence of the O2 absorption at 750 nm becomes significant due to the OMPS's coarser spectral resolution.*
Sentence reformulated as: 'As a consequence of a coarser spectral resolution, the radiance measured at 750 nm is affected by the O2 absorption band. For this reason the OMPS aerosol extinction coefficient retrieval uses the radiance at 869 nm instead of 750 nm, as it was done for SCIAMACHY and OSIRIS.'

   *P16, 8 : with a peak around 35 km → with a worst resolution around 25 km.*
Changed to 'getting worst around 33 km'.

   *Figure: the bottom level of y-axis is marked in all figures.*
If we understand correctly, the reviewer asked us to put a label at the y-axis lower altitude in all plots. This has been done in the revised manuscript.

[revised manuscript text omitted]

---

## Author Response (AR3)

**Replies to Referee #1 on the manuscript 'Retrieval of ozone profiles from OMPS limb scattering observations' by C. Arosio et al.**

We thank the reviewer for the constructive revising process and for the last technical corrections. In the text below, we address the comments from the referee #1. Referee's comments are shown in italicized font and authors' responses are highlighted in blue.

**General comments:**

This paper has been hugely improved through this revising processes. Now, I found this paper to be well written an organized, and the scientific relevance clearly indicated so I would like to give a minor comment prior to publication.

**Minor comments**

 $\rightarrow$  In Figure 10, why the comparison altitude ranges between (a) and (b) is different?

This is related to the convolution procedure: in the case of vertical averaging (panel b), the sonde profile is required to extend at least up to 1.3 km above each coarse grid point. In the case of convolution with AK (panel a), the sonde profile has to extend up to the point where the respective AK approaches zero: accordingly to Fig. 5 panel (a), this happens about 3-4 km above the respective coarse grid point. A sentence has been added in the paper right after the description of the two methods: 'As a consequence, the altitude ranges available for the comparison are different depending on the chosen approach.'.

Page 9 Line 6 : ', that'  $\rightarrow$  eliminating ',' or changing to ', which' might be better

Yes, we changed it to ', which'

Page 12 line 8: a correct description of the aerosol scattering is particularly important.  $\rightarrow$  Please better edit.

Changed to 'it is particularly important to accurately model the aerosol scattering at high northern latitudes...'

Page 13, line 12 Please better revise based on has been used, which was improved in aspect to stray light treatment and pointing corrections in comparison to the previous version.

Reformulated as 'which was improved in terms of stray light treatment and pointing corrections in comparison to the previous version ...'

Page 14, line 8:  $\rightarrow$  doublet and triplet methods, respectively

Done.

Page 14, line  $14 \rightarrow$  Please better revise, it is hard to understand for "With values that follow an approximate linear decrease along the orbit"

We deleted the sentence and added the reference to Sect 2.2 and the Release Notes of L2 data, where the correction is explained.

Page 15, line 2: at these altitudes the ozone concentration and, thus  $\rightarrow$  the ozone concentration

gets very low??

Done: 'in the tropics, where the ozone concentration in this altitude range gets very low.'

Pape 22, line 19: Please edit "was detected, unexpected"

Reformulated as 'was detected, which was unexpected after...'

Page 22: why do you note such like "sonde dataset were retrieved on YYMMDD"?

We replaced 'retrieved' with 'downloaded'. We put the date because the data sets are constantly updated and we used the data available at the beginning of February.

**Retrieval of ozone profiles from OMPS limb scattering observations**

Carlo Arosio1, Alexei Rozanov1, Elizaveta Malinina1, Kai-Uwe Eichmann1, Thomas von Clarmann2, and John P. Burrows1

[revised manuscript text omitted]